# Additively Manufactured Lattice Materials with a Double Level of Gradation: A Comparison of Their Compressive Properties when Fabricated with Material Extrusion and Vat Photopolymerization Processes

**DOI:** 10.3390/ma16020649

**Published:** 2023-01-09

**Authors:** Genaro Rico-Baeza, Enrique Cuan-Urquizo, Gerardo I. Pérez-Soto, Luis A. Alcaraz-Caracheo, Karla A. Camarillo-Gómez

**Affiliations:** 1Tecnológico Nacional de México en Celaya, División de Estudios de Posgrado e Investigación, Celaya, Guanajuato 38010, Mexico; 2Tecnológico de Monterrey, Institute of Advanced Materials for Sustainable Manufacturing, Monterrey, Nuevo León 64849, Mexico; 3Facultad de Ingeniería, Universidad Autónoma de Querétaro, Santiago de Querétaro, Querétaro 76010, Mexico; 4Tecnológico Nacional de México en Celaya, Department of Mechatronics Engineering, Celaya, Guanajuato 38010, Mexico; 5Tecnológico Nacional de México en Celaya, Department of Mechanical Engineering, Celaya, Guanajuato 38010, Mexico

**Keywords:** lattice materials, vat photopolymerization, material extrusion, additive manufacturing

## Abstract

Natural porous materials adjust their resulting mechanical properties by the optimal use of matter and space. When these are produced synthetically, they are known as mechanical metamaterials. This paper adds degrees of tailoring of mechanical properties by producing double levels of gradation in lattice structures via cross-section variation in struts in uniformly periodic lattice structures (UPLS) and layered lattice structures (LLS). These were then additively manufactured via material extrusion (ME) and vat photopolymerization (VP). Their effective mechanical properties under compressive loads were characterized, and their stiffness contrasted with finite element models (FEM). According to the simulation and experimental results, a better correlation was obtained in the structures manufactured via VP than by ME, denoting that printing defects affect the correlation results. The brittle natural behavior of the resin caused a lack of a plateau region in the stress–strain curves for the UPLS structures, as opposed to those fabricated with ME. The LLS increased energy absorption up to 244% and increased the plateau stress up to 100% compared to the UPLS. The results presented in this paper demonstrate that the mechanical properties of lattice structures with the same base topology could be modified by incorporating variations in the strut diameter and then arranging these differently.

## 1. Introduction

Due to their characteristics, mechanical metamaterials are used as light materials with a well-known capacity to absorb energy, high mechanical strength, and stiffness with relatively low density [1,2]. They are often classified into 2D structures (honeycombs), and 3D structures. The latter include: random structures (foams), strut-based lattices, and triply periodic minimal surfaces (TPMS) [3,4,5]. Metamaterials, when properly designed, can have optimal strength-to-weight and stiffness-to-weight ratios; these have inspired the research community to explore uniform or graded topologies. These properties make them potentially applicable to various fields, such as aerospace, biomedical, and transportation [6].

Lattice structures are built from unit cells uniformly distributed within a given volume, known as periodic lattice structures. On the other hand, some structures can also be found with gradual variations of topology [7] or parameters of the unit cell [8]; these are known as functionally graded lattice structures (FGLS) [9]. Other types of topology variation are known as layered lattice structures (LLS); these are structures where variations are defined at different layers of unit cells [10]. 

Lattice structures can also be classified depending on their deformation mechanism, i.e., stretch- or bending-dominated [11]. The characteristic compressive behavior of a lattice structure has three stages according to the standard [12]: (i) initial elastic deformation, (ii) plateau (due to buckling to fracture of the struts, strain is obtained at roughly constant stress), and (iii) densification occurs when all unit cells have collapsed and come in contact with each other [13,14].

Liu et al. [15] created functionally graded porous structures using gyroid and diamond unit cells. The three gradients proposed in [15] were achieved by varying the density, heterostructure, and cell size, achieving comparable mechanical properties to cortical bone. Bai et al. [9], studied additively manufactured FGLS built using selective laser sintering (SLS). The topologies studied were body-centered cubic (BCC) by varying the unit cell size and the diameter of the strut in a unidirectional way. Then, the compression properties were characterized through experimental tests and numerical simulations, with errors of 1.43%, 10.3%, and 8.64%, for stiffness, strength, and plateau stress, respectively. Zhang et al. [16] analyzed structures built with a combination of unit cells (simple cubic and octet) via SLS, including the defects at the interfaces between topologies. The topology variation in the structure allowed it to control its strength, stiffness, and energy absorption. These structures presented instability in their deformation due to the connection interfaces of the gradients and resulted to be more flexible than periodical structures. Dumas et al. [17] developed a graded cell structure applying a scale factor in the nodes of the diamond unit cell and studied its compression response both numerically and experimentally. When comparing the experimental results with the simulation results, they present a divergence of 40% for stiffness and 50% for yield strength. Li et al. [18] studied (theoretically and experimentally) the compression properties of lattice sandwich structures with variable cross-sections. The structures were manufactured using stereolithography (SLA), and the strength of the structures with variable cross-sections was greater than structures with uniform cross-sections. These works have covered the characterization of the mechanical properties in structures with variations of the topology, unit cell size, or relative density; however, they do not explore the feature of varying the geometry of the strut cross-section, which could be an additional parameter to study towards the tailoring of the effective properties of porous media.

A common challenge to overcome in lattice structures with a geometric variation of the struts is how these are connected to adjacent struts with different cross-sections, which leads to reduced loading bearing capacity. To overcome this, Goel et al. [19] proposed a methodology for the design of B-spline connections for adjacent struts with different diameters in BCC unit cells; via numerical analysis, it was shown that functionally graded structures with B-spline connections are more rigid than uniformly periodic lattice structures (UPLS).

FGLS have been explored to mimic the bone structure, density, and stiffness of trabecular or cancellous bone, often needed in prosthesis design to allow bone growth and to fix the implant naturally with human bone [20,21]. Wang et al. [22] used a graded octet cell structure to design an acetabular component to improve implant-bone stability; the graded cell structure allowed the elastic modulus of the implant to be matched with that of bone. Alkhatib et al. [23] studied the load transfer towards the femur from the stem of a hip prosthesis built with periodic and graded structures using BCC unit cells, discovering that stems with graduated structures transfer a more significant load to the femur than stems with structures uniformly periodic. The implementation of lattice structures to design implants such as hip prostheses could be a potential application of lattice structures proposed in this work, as here, an additional level of parameter variation is presented, aiming to achieve a wider range of mechanical properties.

Additive manufacturing (AM) is the most frequently used method to fabricate lattice structures since their designs demand complex geometries that are impossible or difficult to make using conventional manufacturing methods [24,25,26]. The printing process parameters and defects influence the lattice structures’ mechanical behavior. Guerra Silva et al. [27] determined that the layer height and the material supplier are the two most influential parameters that affect plateau stress. Still, the topology of the unit cell and the size of the struts have a more significant impact on the mechanical properties.

The review of related literature brings up several lines that still deserve further research. As seen, AM appears as the ideal fabrication technology for the fabrication of these structures; however, at relatively small dimensions, the numerous defects inherent to it demand attention. Additionally, intending to produce structures with tailor-made properties, different attempts use variations of the unit cell distribution or changes in the topology. Works that have attempted to vary the mechanical properties of the lattice structures at the strut level are scarce. Hence, the main objective of this research was to create new layered lattice structures using body-centered cubic-type unit cells (BBC) with different cross-section variations of the struts. The characterization of the mechanical properties is carried out via numerical simulations and laboratory experiments on additively manufactured samples. Specifically, fused filament fabrication (FFF) and liquid crystal display (LCD) methods were employed, which are classified as material extrusion and vat photopolymerization techniques according to [28], respectively.

## 2. Materials and Methods

The general methodology followed in this work is presented in Figure 1. Here, lattice structures were computationally modeled, then these models were additively manufactured and further analyzed via finite element simulations. The parametrization and unit cell definition, along with the manufacturing simulation and characterization parameters, are given in this section.

### 2.1. Parametrization of Lattice Structures and Computational Modeling

This work aims to tailor the mechanical properties of lattice structures at two different levels: (i) functionally graded modification of the strut cross-section and (ii) layered distribution of unit cells. The lattice structures are built with unit cells similar to the BCC. The angle of the inclined struts was modified from 45° to 30° as a result of the strut cross-section variation; struts overlap at locations near the central node of the unit cells, and by changing the inclined angle to 30° this is avoided. The structures were modeled in Solidworks^®^ (v2020, Dassault Systèmes, Waltham, MA, USA), using unit cells with dimensions of 8×8×5 mm, and a nominal volume $V_{n}=320$ mm^3^. The lattice structures are comprised of unit cell arrangements of 5×5 in the XZ-plane and 10 unit cells on the Y-axis, as shown in Figure 2. The dimensions of the unit cells were selected considering the resolution of the 3D printers used to manufacture the samples.

The first level, with the aim of tailoring the mechanical properties of lattice structures, is to grade varying strut cross-sections along their length. Six cross-section variations were considered: cosine (COS), cosine 2 (COS2), double slope (DS), double inverted slope (DIS), positive slope (PS), and negative slope (NS). This mathematical parameterization is proposed to vary the diameter of the struts in the unit cell achieving control of the accumulation of material in the center and at the ends of the strut’s length. These variations are plotted in Figure 3, along with the resulting unit cells. The corresponding lattice structures formed with these unit cells are then depicted in Figure 4 (5×5×10 unit cells). The relative density $\bar{\rho }$ of the lattice structures was calculated using the volume fraction as: $\bar{\rho }=V_{e}/V_{s}$, where $V_{e}$ is the volume of the actual space occupied by the lattice structure, and $V_{s}$ is the overall volume of the lattice structure, including the empty spaces.

The second level controls the unit cell distributions layer-by-layer, i.e., layered lattice structures (LLS). Six LLS were designed considering an initial finite element simulation of the structures under compressive loading shown in Figure 4. From these simulations, their stiffness was measured so that these were labeled as lower elastic modulus (LM), intermediate elastic modulus (IM), and higher elastic modulus (HM). The lattice samples were divided into three zones (bottom, middle and top) to accommodate the unit cells with different elastic moduli, as illustrated in Figure 5. The number of unit cell layers in the zones mentioned above is adjusted for each proposed structure to match their relative densities. Figure 5 shows a labeling scheme used for the LLS arrangements studied: HM-LM-HM stands for a structure with a low elastic modulus in the middle, and high elastic modulus in the top and bottom zones. The remaining structures studied have the following arrangements: LM-HM-LM, HM-IM-LM, LM-IM-HM, LM-IM-LM, and IM-LM-IM. The arrangements were proposed in this way to obtain structures with three zones of different stiffness and in turn avoid having the same stiffness in consecutive zones.

### 2.2. Parametrization of Lattice Structures and Computational Modeling

ANSYS^®^ (Canonsburg, PA, USA) software was used to perform the quasi-static and linear compression simulations. Fixed support was placed on the lower plate of the cell structure to simulate the conditions of the experimental tests, and a displacement in the Y-axis was applied to the upper plate (see Figure 6). The mesh of the model was made with tetrahedron solid elements of 0.4  mm in size.

The effective elastic modulus of lattice structures was obtained by: $\bar{E}=FL/\bar{A}\delta$, where F is the force applied to the structure, L is the original length of the lattice structure in the loading direction, $\bar{A}$ is the apparent area where the force is applied, and δ is the displacement of the structure due to the applied load. The displacement of the structure was obtained from the FE simulations after the load was applied and thus obtaining the apparent elastic modulus $\left(\bar{E}\right)$. 

### 2.3. Additive Manufacturing of Lattice Structures

The lattice structures were additively manufactured using two printing technologies: FFF and LCD; a Creality^®^ (Shenzhen, China) Ender 3D printer was used for FFF, and a Creality^®^ LD002-R printer for LCD. Computational models (STL files) for the FFF method were processed in Ultimaker Cura^®^ V4.8.0 (Ultimaker B.V., Utrecht, The Netherlands) software, and PLA material supplier (in the form of filament) was used as raw material. To determine the manufacturing parameters for the FFF process, tests were carried out with different sizes of unit cells, nozzles, and printing speed; in addition, a displacement in the Z-direction of 0.1 mm was set to the printing head, when it moved from one point to another. Finally, the minimum size of the unit cell that could be printed is shown in Section 2.1 using the following parameters: extruder nose diameter of 0.2 mm, layer height of 0.1 mm, 100% infill, bed temperature of 70 °C, extruder temperature of 180 °C and printing speed 40 mm/s. On the other hand, models (STL files) for LCD were processed using Chitubox^®^ (Guangdong, China) V1.6.2 software; samples were made with Creality^®^ photosensitive resin. The layer thickness was 0.05 mm; the exposure time was 9 s and the exposure time for the first layer was 50 s. Isopropyl alcohol was used to remove the excess liquid resin from the samples. The samples were then cured by exposing them to ultraviolet light for 10 min.

In both manufacturing processes, the printing parameters were not optimized, i.e., standard parameters were used. The printing parameters were kept constant for all samples and printed without supports, and no special methods were required for adhesion to the print bed. All samples were fabricated-oriented, so the XZ-plane was parallel of the printing platform. The weight of the samples was measured with a RADWAG AS 220.RS balance to subsequently calculate the relative density.

#### Micrograph Characterization of Additively Manufactured Lattices

The UPLS fabricated by FFF and LCD were further inspected using a ZEIN microscope to contrast them with their computational model versions. The length of the spaces between unit cells were measured, i.e., horizontal (1) and vertical (2) directions, the length of the struts (3), and their diameter in three regions, i.e., at the end (4), at the center (5) and the beginning (6) of the strut were measured. Figure 7 shows the regions where the struts were measured.

### 2.4. Mechanical Characterization of the Constituent Additively Manufactured Base Materials

The mechanical properties of the base materials used to fabricate the lattice structures were characterized by quasi-static tensile tests on solid dog bone specimens manufactured using the same two printing processes mentioned in Section 2.3. The shape and dimensions of the tensile samples were obtained from the standard in [29]. The samples were printed in the same orientation in which the struts are encountered in the unit cells (30°) and with the same printing parameters. Tensile tests were performed with a ZWICK^®^ Z-050 machine at room temperature; the test speed was 5 mm/min. The bulk density of the base materials was calculated by weighing a fully filled printed 1 cm^3^ cube. The density of the materials obtained was 1.24 gr/cm^3^ and 1.184 gr/cm^3^ for PLA and resin, respectively.

### 2.5. Experimental Characterization of the Compressive Response of Additively Manufactured Lattice Structures

Quasi-static compression tests were carried out to evaluate the mechanical properties of the designed structures with a ZWICK^®^ Z-050 machine at room temperature. The test speed was 5 mm/min, using two plates with a diameter of 150 mm; also, as shown in Figure 8, a NIKON^®^ (Melville, NY, USA) P7000 camera was used to capture the deformation of the lattice structure during the test.

## 3. Results and Discussion

This section presents and discusses the manufacturing defects of the lattice structures, the properties characterized by the compression experiments, and the comparison of stiffness determined by simulations and experiments.

### 3.1. Additive Manufacturing Defects in Uniformly Periodic Lattice Structures

The FFF fabrication took up to 23 h for each structure, while the simultaneous fabrication of two structures took up to 6 h with LCD. The main parameters that allowed the fabrication of the structures in FFF were the diameter of the nozzle of 0.2 mm and the printing speed of 40 mm/s. Because the minimum diameter of the struts at their thinnest area was 0.4 mm, printing with a larger diameter nozzle was not possible. The maximum speed that could be applied was 40 mm/s because if this parameter was increased, the struts of the COS2 and DIS structures would bent, due to being thin in the center of their length. On the other hand, the speed did not decrease further because the printing time increased considerably.

Two measurements were taken for each parameter, as shown in Table 1, and the average was calculated (see Table 2). Compared with the CAD models, the vertical and horizontal distances resulted in greater differences in the samples printed by FFF due to the accumulation of material in the connecting zone between unit cells and at the midpoint of the strut lengths, respectively. 

The samples manufactured by FFF had a more significant difference in the diameter measured at the midpoint of the strut length because the COS2 and DIS structures were thinner in this area. Also, unbonded material was observed at these zones; this can be seen in the third and fifth rows of the third column in Table 1. On the other hand, in those models with thicker mid-sections of the strut length, an accumulation of material was observed. LCD-printed COS2, DS, and DIS structures showed more significant differences in strut length; this can be seen in the third, fourth, and fifth rows of the fourth column in Table 1.

The structures made with FFF presented defects such as threads between adjacent unit cells in the PS, NS, DIS, and COS2 structures, the latter resulting in the highest number of threads. These defects can be seen in the third column in Table 1. In some regions along the struts, there were drop-shaped detachments of material; this defect can be seen in the fifth and sixth row of the third column in Table 1. In the third column, in the third, fifth, sixth, and seventh row in Table 1, it is shown that clumps of material were formed at the intersection of the struts within the unit cell. In the third column, in the third and fifth rows in Table 1, it is shown that the COS2 and DIS structures were the ones that presented defects at the struts’ midpoints, as these were thinner in this area where drops of material were formed mainly due to the lack of supports in the printing process.

The relative density $\bar{\rho }$, was measured from triplicates of each structure. These were weighed, and considering the density of the material, the average volume of the lattice structures was calculated as $V_{s}$. Then this volume was divided by the volume of occupied space, $V_{e}$, to obtain the actual volume fraction. Again, when comparing with the CAD version of each sample, the structures with the most significant difference in relative density between the CAD and FFF models were COS2 and DIS, with a difference of 4.6% and 4.65%, respectively; this can be seen in Figure 9. The printed structures had higher relative density than the computational models in the rows corresponding to the COS2 and DIS structures measurements in Table 2. The vertical distance measured in the space between unit cells is lower in the printed samples than in the computational models, denoting the existence of more material in this area and, therefore, higher relative density. 

The structures manufactured in LCD resulted in a relative density closer to that of the CAD model, as shown in Figure 9a. The FFF structures had considerably noticeable defects (accumulation of material in the connections and the narrow areas of the struts), and due to these defects inherent to the printing process showed a more significant discrepancy in relative density concerning computational models. Additionally, the apparent porosity $\left(1-\bar{\rho }\right)$ is calculated and plotted in Figure 9b.

### 3.2. Apparent Compressive Stiffness via FEA for the Assembly of the Layered Lattice Structures

As mentioned in Section 2.2, initial FEA simulations were carried out to characterize the apparent stiffness modulus $\left(\bar{E}\right)$ of the lattice structures. All FE models were fed with the base material properties obtained from the dog bone samples testing (Section 2.4); Young’s modulus for PLA and resin is 2.2±0.023 GPa and 1.56±0.145 GPa, respectively. A comparison of this apparent elastic modulus is normalized with Young’s modulus of the parent material E and is presented in Figure 10. These results allowed the classification of the lattices accordingly: NS as HM, DS as IM, and COS as LM. These were used to build the lattice structures with the methodology depicted in Figure 5, leading to the structures presented in Figure 11.

### 3.3. Additive Manufacturing Layered Lattice Structures

Triplicates of the LLS shown in Figure 11 were additively manufactured with FFF and LCD. These were also analyzed with the microscope, focusing on the transition zones. These zones are the transitions between HM, IM, and LM structures. These micrographs are presented in Figure 12 and Figure 13 for FFF and LCD, respectively.

Significant defects were observed in the zones between unit cells with struts with different cross-sections. These were due to sudden changes in the geometry of the struts, mainly encountered in the transitions between NS and COS. In this case, the connection between struts was lost entirely (Figure 14a). Another defect was that some intended struts or unit cells were not fabricated (Figure 14b,c). The material intended for these was accumulated in nearby regions (Figure 14d). LCD-printed structures were flawless, and the connecting interfaces had a good bond between the struts of the unit cells. This can be seen when comparing Figure 14a,d with Figure 14e.

A comparison of the intended (CAD) relative densities and apparent porosities with those measured on LLS fabricated via FFF and LCD is presented in Figure 15. Note that differences between relative densities of CAD models and FFF samples are negligible. This is mainly due to the absence of material at the regions where there is a change of strut cross-section variation in adjacent unit cells (see Figure 14a,d). On the other hand, differences in relative density between CAD models and LCD samples are more noticeable due to a combination of aspects. Firstly, LCD samples result with minimum defects at these regions where unit cells with different strut cross-section variation are adjacent (Figure 14e); secondly, struts in LCD samples are thicker than the CAD model ones. It is also important to highlight that, among the CAD models and all the different strut cross-section variations the maximum difference was 1.56%.

### 3.4. Mechanical Properties of Lattices Structures under Compression

A total of 72 samples were tested under compressive loads. These include three replicas for each design for each AM process used. The results include the characterization of several properties including stiffness, peak and plateau stress, and the energy absorbed.

#### 3.4.1. Characterization of the Mechanical Properties of FFF Structures

The average load-displacement data of the UPLS structures manufactured with PLA is shown in Figure 16. UPLS structures manufactured via the FFF process resulted in brittle fractures. These fractures followed an oscillating plateau region. This is also related to a layer-by-layer rupture which agrees with the reported results [9,30].

In Figure 16a, the maximum load for the COS lattice structure occurred at a displacement of approximately 2.5 mm; this structure started to fail at its central layer with the collapse of the unit cells, the central unit cells began to collapse (see Figure 17), and the upper and lower cells were still intact at the beginning of densification.

In Figure 16b, the maximum load in the COS2 lattice structure occurred at a displacement of 1.37 mm; later, the upper layers began to collapse. At a displacement of 15 mm, a diagonal was formed in the upper part of the structure due to the failure of some unit cells (see Figure 17); the surrounding unit cells below these layers began to compress as stress increased, and when densification began, the unit cells at the bottom of the structure had not yet collapsed.

Load-displacement curve (Figure 16c) for the DS structure revealed that at a displacement of 15 mm the central unit cells collapsed diagonally and the unit cells located in the center of the structure had collapsed entirely meanwhile the upper and lower layers remained almost undeformed (see Figure 17). 

The load-displacement curve for the DIS lattice structure, Figure 16d, shows that the unit cells of the central layer collapsed at 15 mm. Half of the structure had been compressed and collapsed, and a diagonal was formed of 45° that divides the collapsed zone and the zone with roughly undeformed unit cells (see Figure 17). At the beginning of the densification, most of the unit cells had been compressed.

When the PS structure had reached its peak force (see Figure 16e), slight bending of the struts in the unit cells was observed. At 15 mm of displacement, some struts collapsed at the center of the structure (see Figure 17). At 30 mm displacement, the structure has failed at its bottom zone (see Figure 17).

For the NS structure (Figure 16f), the peak force occurred at 17.4 mm. Subsequently, detachment of the vertical row happened on the right part of the structure, and the rest of the vertical rows had displaced laterally, resembling a failure due to buckling in the structure (see Figure 17). Again, the unit cells at the top and bottom of the structure did not collapse when densification occurred.

The cross-section variation in the unit cells influences the failure form of the structures, the magnitude of the maximum stress, and the plateau stress; the densification occurred approximately at 30 mm in all the structures; in addition, the upper and lower cells were not damaged, this effect is attributed to the fact that these cells were attached to the plates, and these provided greater stiffness to these cells. The data for the maximum load, the displacement where the maximum load occurred and the magnitude of the load at the moment of densification are summarized in Table 3.

#### 3.4.2. Characterization of the Mechanical Properties of LCD Structures

Figure 18 shows the force-displacement graphs of the UPLS structures manufactured via LCD. The structures manufactured with photosensitive resin did not present a plateau or densification; the maximum load of the structures occurred at a displacement of 3 mm to 6 mm; after the maximum load occurred, it fell abruptly due to the sudden detachment of the unit cells or fragments of the structure. This failure mode can be attributed mainly to the brittleness of the material. Figure 17 shows the deformation of the structures during the compression test. It was not possible to analyze the PLA and resin structures at the same range of displacement because the UPLS manufactured via LCD broke between 3 mm and 6 mm of displacement, while those of PLA their rupture occurred at approximately 30 mm of displacement.

According to Figure 18a in the COS lattice structure, the maximum load occurred at 3.2 mm of displacement, and then the load dropped abruptly due to the detachment of some unit cells, forming a 45° diagonal structure (see Figure 17). Figure 18c shows that the DS lattice structure presented a behavior similar to that of the COS lattice structure; after the maximum load, there was a drop in load capacity due to the detachment of the unit cells diagonally at 45° (see Figure 17). After the maximum load in the COS2 lattice structure is reached (Figure 18b), a load drop occurred due to the rupture of the lower unit cells. A 45° diagonal was formed in the lower part of the structure at 3.2 mm displacement, and also a lateral displacement of the lower part of the structure was observed. This is where the rupture occurred (see Figure 17). In the DIS lattice structure, Figure 18d, the maximum load occurred at a displacement of 3.2 mm; failure was sudden due to the detachment of the lower part of the structure at 45° (see Figure 17).

Figure 18e shows that the PS lattice structure presented its maximum load at 5 mm of displacement. At 9 mm of displacement, the detachment of the unit cells was observed diagonally at 45° (see Figure 17). Figure 18f shows that the maximum load in the NS reticular structure occurred at a displacement of 4.9 mm, where the deformation of the struts and detachment of some unit cells in the vertical rows on the left side of the structure was observed. At a displacement of 15 mm, most unit cells were detached and only those in the center of the structure remained (see Figure 17); this failure mode also occurred in the NS lattice structure manufactured via FFF.

Although the structures were built with the same topology (BCC), the proposed cross-sectional variations influence the failure mode of the structure; this can be seen in Figure 17. As explained above, the brittle behavior of the resin structures did not allow the structure to have a plateau and did not densify either; however, it can be seen that there was a similarity in the failure between the PLA and resin structures. For the COS structure, a 45° diagonal was observed, formed by the rupture of the central cells in the PLA structure, and in resin, the same diagonal was formed by the detachment of the cells in the lower part of the structure; this failure was similar in DS, but the diagonal was to the left in both structures (PLA and resin). In COS2, failure started in the cells at the top and the bottom of the structure for PLA and resin, respectively. In the case of DIS, the form of failure was different in the structures, while for PLA, the struts were bent, allowing the layers of the structure to join. In resin, there was a detachment of the unit cells in the lower part of the structure. It should be noted that the DIS structure is one of those that presented the most significant defects in the FFF manufacturing process because the geometry of the strut was not clearly defined, specifically in its central part. They showed a lack of material and weak joints. The difference in the form of failure can be attributed to printing defects. For PS in both structures, a 45° diagonal was formed, and for NS, some vertical rows were detached in both structures, resembling a buckling failure.

The load capacity in most structures manufactured by LCD was more significant than the FFF. The most remarkable difference occurred in the COS2 and DIS structures with a difference of 75% and 87%, respectively. This is mainly due to the defects that occurred in the COS2 and DIS structures manufactured by FFF. The COS and PS structures fabricated via LCD had higher load capacities than the structures fabricated via FFF, with a difference of 18% and 48%, respectively. The DS and NS structures manufactured by FFF had a higher load capacity than those manufactured by LCD.

#### 3.4.3. Maximum Stress, Plateau, and Energy Absorption

The average force obtained from the three samples of each structure and its apparent area of 40 mm^2^ (area of the top plate) were used to calculate the maximum and plateau stress. These are presented in Figure 19a,b, respectively. The COS2, DIS, and NS structures made of resin resulted in higher yield strengths; the struts of these unit cells were thicker in the central part of the unit cell, giving greater strength to the structure.

Differences in the trends observed in Figure 19a between FFF and LCD lattice structures were due to COS2 and DIS structures manufactured by FFF resulting in the most significant defects, such as a lack of material in the narrow area of the strut, and the geometry of the strut was not defined correctly, which influences its strength, as it is shown in Table 1.

The maximum stress for the COS and DS structures were similar in PLA and resin according to Figure 19a; Table 1 shows that these structures had more defects when fabricated via FFF. Porosity was lower in the PLA structures than in resin ones, compensating for the existing defects in these structures. For the NS structure, there was no considerable difference in the magnitudes of the struts between the PLA and resin structures, as shown in Table 2.

Figure 19b shows the plateau stress for the structures printed via FFF; it can be seen that COS, DS, NS, and PS resulted in similar plateau stress between 0.1291 MPa and 0.1708 MPa. The COS2 and DIS structures were the ones that presented the lowest plateau stress. 

These structures were the weakest when made of PLA because the geometry of the strut was narrow in the center, coupled with the fact that these structures presented printing defects that compromised their structural performance.

Results according to the stiffness obtained by FEA and experimental compression tests are shown in Figure 19c,d, respectively. A better correlation of results in the resin structures was observed. The maximum difference between the simulation and experimental results in the DS structure manufactured in resin was 28.78% and in PLA 41.62%, because the manufacturing process (LCD) allowed the samples to be manufactured with greater precision. Note in Figure 19d that in some of the structures, the stiffness was higher in the experiments than in the simulation models; this effect was attributed to the fact that the resin structures have a higher relative density because the struts were larger than in the CAD models.

The COS2 and DIS structures manufactured via FFF were the weakest among the rest, which can be attributed to the defects present in the structure. However, they were among the structures that presented higher stiffness in both manufacturing processes; the common characteristic that these structures had is that the struts had a larger diameter in the center of the unit cell, which provides higher stiffness to the structure.

As shown in Figure 9, the structures with higher relative density were COS and DS, resulting in lower stiffness than COS2 and DIS. This in turn suggests that the stiffness is related to the geometry of the strut cross-section, not only to their relative density. The stiffness determined via FEA shows the same tendency in PLA and resin; the experimental results with PLA presented a discrepancy with the simulation results, mainly in the COS2 and DIS structures where the stiffness determined by simulation was higher than the experimental one, according to the Figure 19c. This was mainly attributed to the defects that occurred at the central and narrow part of the strut. Although there was no significant difference in the dimension of this zone of the strut between the CAD model and the printed one (FFF), it can be observed that the material was not entirely bonded in this area, which compromises its structural integrity. Another structure that presented a significant discrepancy was DS, although in this case, the experimental stiffness was greater than that of the simulation; according to the measurements of the effects, the space without material was smaller in the samples made in PLA than in the CAD models, and due to the existence of more material the stiffness increases for this structure.

The absorption energy was determined in the plastic zone because the elastic deformation provides recoverable energy absorption; in the densification zone, the structure has been completely compressed, and in this zone, there is no energy absorption [31]. For UPLS and LLS, the absorption energy was determined by the area under the curve after the elastic zone and until the beginning of the densification. Because the structures fabricated via LCD did not present a plateau, the energy absorption in these structures was minimal, as most of it was lost in fracture. Figure 19e shows that the structures with the lowest energy absorption were COS2 and DIS; due to their low strength and the fact that the plateau stress was lower than the rest of the structures, the strength and the plateau stress, and the energy absorption were affected by the geometry of the strut, mainly in its central shape.

#### 3.4.4. Mechanical Properties of Layered Lattice Structures under Compression

In the load-displacement graphs of the LLS shown in Figure 20, staggering was observed at the plateau because the low-intermediate-high zones, being constructed with different unit cells, failed and densified at different magnitudes of displacement. Because the unit cells, with which the LLS were built, had different stiffnesses and strengths, the structures presented different forms of deformation. The area where COS was found deforms at a lower deformation rate than DS and NS due to its lower stiffness; the area where NS was found deforms at higher magnitudes of displacement as it was the stiffest topology.

In the structures manufactured with FFF and LCD, failure started in the area where COS was located because it had the lowest strength. The structures in Figure 20a,b contain NS, the maximum load (524.6 and 642.4 N, respectively) in these structures occurred at 30 mm displacement, COS was completely densified, and NS began to deform (see Figure 21). HM-IM-LM (Figure 20c) and LM-IM-HM (Figure 20d) structures started to fail in the COS zone followed by DS, at 30 mm these unit cells densified and NS began to deform (see Figure 21). The LM-IM-LM structure is shown in Figure 20e; in this structure the COS unit cells were compressed to 15 mm displacement, and the structure had a more uniform deformation than the rest of the structures (see Figure 21). The central zone of the IM-LM-IM structure started to deform at 15 mm (see Figure 20f), and at 30 mm of displacement this zone was completely densified.

Figure 22 shows the LLS manufactured by LCD; it is observed that these structures suddenly broke in the area where the layers with COS unit cells were located. Failure of HM-LM-HM structures occurred at the interface of changing unit cells COS to NS (see Figure 21).

Figure 22b shows the force-displacement graph of the LM-HM-LM structure. At maximum load there was COS unit cell detachment and failure continued in the transition interface between COS and NS (see Figure 21). The only structure that did not present densification was HM-IM-LM, this structure combined the unit cells of NS-DS-COS. Because the COS unit cells were at the top of the structure and broke suddenly, the top of the structure shifted to one side and the load dropped abruptly (see Figure 21).

Figure 22d shows the force-displacement graph of the LM-IM-HM structure. Failure in this structure occurred at its lower zone where the COS unit cell was located. The failure continued with DS unit cells, and this allowed the structure to densify (see Figure 21). In the LM-IM-LM (Figure 22e) and IM-LM-IM (Figure 22f) structures, failure was similar: a diagonal at the central part of the structure was formed (see Figure 21).

The plateau of the structures had small steps due to the rupture of the zones of the structure at different magnitudes of displacement, and the densification of the structures occurred approximately at 35 mm of displacement. Two slopes were observed in the elastic zone, and the simulation prediction resembles the first curve. This zone was considered in the comparison between the simulation and experimental results, presented in Figure 23c and 23d, respectively.

Figure 21 shows the deformation of LLS manufactured by FFF and LCD; due to the brittleness of the resin structures, its failure was sudden, breaking in the layers where COS unit cells were placed at approximately 20 mm displacement. In the LM-HM-LM and LM-IM-LM structures, a 45° diagonal was formed when the COS cells were broken. The IM-LM-IM structure has COS unit cells in its central part, and a diagonal was observed at the fractured moment of these unit cells. With the integration of unit cells with struts with different cross-sections, the lattice structures reached densification at roughly 30 and 35 mm of displacement, except for HM-IM-LM, which has a COS structure only in the upper part. The shape of fracture of the resin and PLA structures differed; however, the failure of the structures with both manufacturing processes occurred in the layers with COS topology (topology with less stiffness and strength).

Deformation mechanisms were better appreciated for PLA structures. The COS unit cell layers deformed majorly, and the struts of the unit cell bent. These COS layers were also the ones that showed densification before any other strut cross-section variation. This was due to the low strength of COS compared to the other cells used in the structures. In the structures that combine the COS and DS cells, the fracture of the struts starts at the joining zone of these cells; later, the COS zone densifies, followed by DS. For the structures that combine the three-unit cells, it was observed that COS and DS densified entirely, and subsequently, NS started to deform; just like COS and DS, the struts of NS flexed, showing that the deformation mechanism was dominated by bending.

Figure 19a shows that the NS structure made of PLA was the strongest, approximately 0.9 MPa, while COS and DS have a strength of 0.2 MPa; LLS structures made of PLA were more robust than the UPLS due to the strength provided by NS. LLS manufactured with resin had strength similar to the UPLS, including HM-LM-HM and LM-HM-LM, which were less resistant than the COS, DS, and NS structures (see Figure 23a). However, significant variability was observed in the experimental results. Data for load capacity, displacement, and densification load are summarized in Table 4 for the LLS structures.

As mentioned above, the plateau in the LLS presented staggering due to the fracture of the zones of the structure at different magnitudes of displacement. COS, DS, and NS structures presented plateau stress of 0.1291 MPa, 0.1497 MPa, and 0.1708 MPa, respectively, while for the LLS structures, their plateau stress was between 0.1739 MPa to 0.3518 MPa, as shown in Figure 23b, which indicates an increase when unit cells with different stiffness and strength were combined. For the resin structures, the combination of COS, DS, and NS allowed for a plateau during the deformation and densification in most of the LLS; due to the rupture by zones of the structure, the LLS that did not present a plateau or densification was HM-IM-LM because only the upper part of the structure has the COS variation.

The distribution of the unit cells in LLS was relevant to their stiffness; even though LLS were designed with the same relative density, the simulation results showed that placing layers of unit cells with lower stiffness in the center of the structure decreases the stiffness of the structure when compared to the rest of the models. The experimental stiffness results of the resin compression tests correlated with the simulation results are presented in Figure 23d. It is worth mentioning that the experimental stiffness was higher than the simulation one because, in the printed samples, the struts of the unit cells were larger than in the CAD models, a phenomenon inherent to the manufacturing process. Despite this, the experimental results showed the same trend; the HM-LM-HM, and IM-LM-IM structures were the ones with the low stiffness, as shown in Figure 23d. These structures had the COS structure in their central part, which decreases their stiffness. The stiffest structure was LM-HM-LM, which concentrated NS in its intermediate part, indicating that the stiffness of the LLS depends partially on the stiffness of the structure located in its central part. The trend in the structures manufactured via FFF was different from those of resin and simulation (Figure 23c); this effect arose due to manufacturing defects. According to Table 2, the defects in COS and DS made the strut thicker; this effect can also be verified in Figure 9, where COS, DS, and NS had higher relative density than the CAD models, so the discrepancy between the structures made with PLA and the simulation is primarily attributed to manufacturing defects.

Figure 19e shows that in the UPLS, the structure with the highest energy absorption was DS with 0.109 J/cm^3^. In the LLS, the energy absorption increased except in the LM-IM-LM and IM-LM-IM structures that absorbed the same energy as DS; LM-HM-LM is the structure that most increased the energy absorption up to 244% when compared to DS. Recall from the tests on resin UPLS that almost no plateau region was obtained. Once the structures were not formed from the same unit cell, but through the variations explained, the resin LLS did present plateau stress, as shown in Figure 22. 

## 4. Conclusions

The UPLS and LLS were printed via FFF and LCD without supports; the structures manufactured by FFF were those that presented defects such as accumulation of material in the nodes and the narrow areas of the struts. The most significant defects were presented in the LLS built with NS and COS unit cells because there was no connection between the struts of these structures. This defect could be corrected by smoothing the geometry change at the connection of these unit cells.

The UPLS printed via LCD had a brittle behavior that did not allow the structure to present plateau or densification during its deformation, so it was not possible to measure the energy absorption. However, the lattice structures printed by this method presented a good correlation between the simulation and the experimental stiffness. The DS structure was the structure with the most significant discrepancy between the experimental and simulation stiffness with 28.78%. Although the UPLS printed via FFF had defects, there was also a correlation between the stiffness of the experimental simulation; however, the defects presented in the LLS manufactured with PLA did not allow a correlation between the simulation and experimental stiffness to exist.

In the geometric characterization, it was observed that the struts of the unit cells manufactured by both processes were more significant than the CAD models; the manufactured structures had a higher relative density than the CAD models.

The change of cross-section in the struts was the first level of gradation proposed, which allowed modifying the strength, stiffness, plateau stress, and energy absorption of the BCC structures. Lattice structures with thick struts in the center of the unit cell were stiffer and stronger. This trend was observed in structures made with resin and simulations; structures manufactured through FFF did not have this tendency due to manufacturing defects that compromised structural performance.

The second level of gradation was implemented with LLS, and the experimental results showed an increase in the strength of the lattice structures in both fabrication processes. Furthermore, the plateau stress was higher for LLS than for UPLS. In resin, a comparison could not be made due to the absence of a plateau in the UPLS; however, the resin LLS presented a plateau during the deformation of the structure due to the rupture of the three zones of the structure at different levels of deformation. 

Although LLS had the same relative density, the stiffness of these structures depended on the arrangement of the unit cells. The lattice structures that had unit cells with low stiffness in their central zone were the least rigid. Energy absorption and plateau stress increased 244% and 100% in LLS compared to the UPLS; in the deformation of LLS, it was observed that the plateau presented steps because the COS, DS, and NS unit cells deform at different levels of deformation; this allowed the structure to absorb more energy.

In general, the mechanical properties studied in this work depend on the geometry of the strut, being higher when the strut has a wider cross-section near the center of the unit cell. Additionally, it was demonstrated that with LLS structures, the right selection of unit cells in each layer could result in structural modifications, not only of the effective properties but also of the failure mechanisms. It is important to highlight that this was demonstrated here for the BCC structures, and the effect on lattices built from other topologies, or using different topologies within the same sample, is still an open issue. 

## Figures and Tables

**Figure 1 materials-16-00649-f001:**
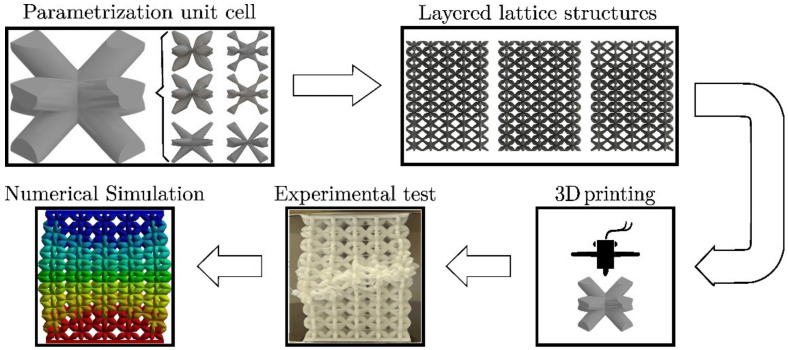
The methodology followed: parametrization of the unit cell, design of layered lattice structures (LLS), 3D printing, experimental test, and numerical simulation.

**Figure 2 materials-16-00649-f002:**
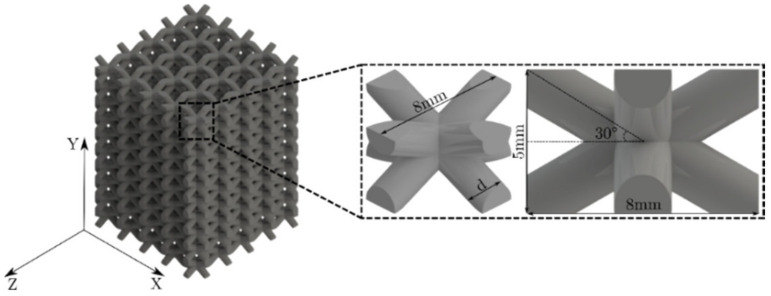
CAD model of the lattice structure and unit cell dimensions.

**Figure 3 materials-16-00649-f003:**
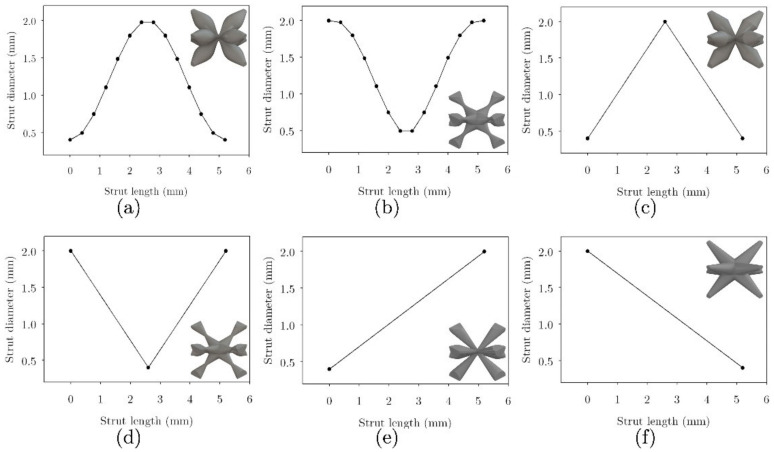
Cross-section variation: Type of variation (**a**) Cosine (COS), (**b**) Cosine2 (COS2), (**c**) Double Slope (DS), (**d**) Double Inverted Slope (DIS), (**e**) Negative Slope (NS) and (**f**) Positive Slope (PS).

**Figure 4 materials-16-00649-f004:**
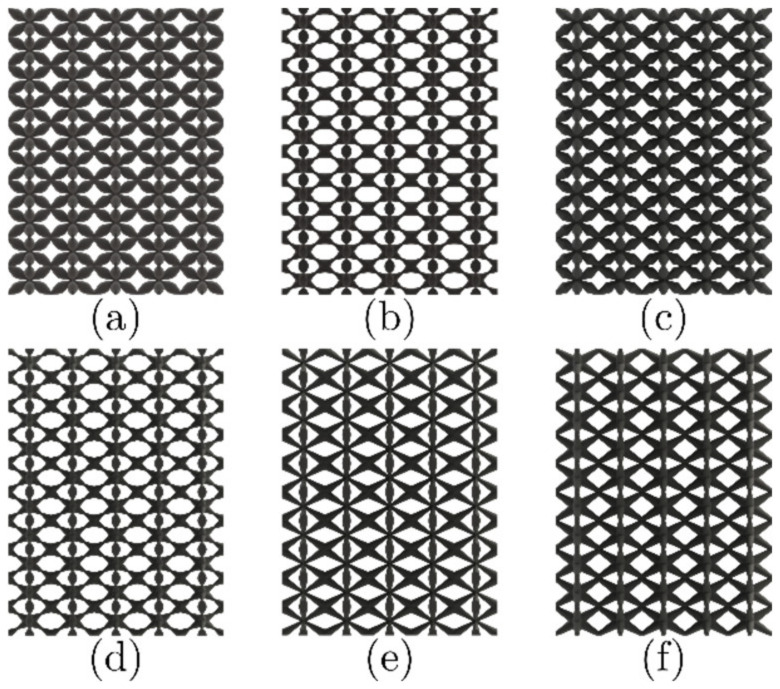
Lattice structures with cross-section variation of the strut. Type of variation (**a**) COS, (**b**) COS2, (**c**) DS, (**d**) DIS, (**e**) PS, and (**f**) NS.

**Figure 5 materials-16-00649-f005:**
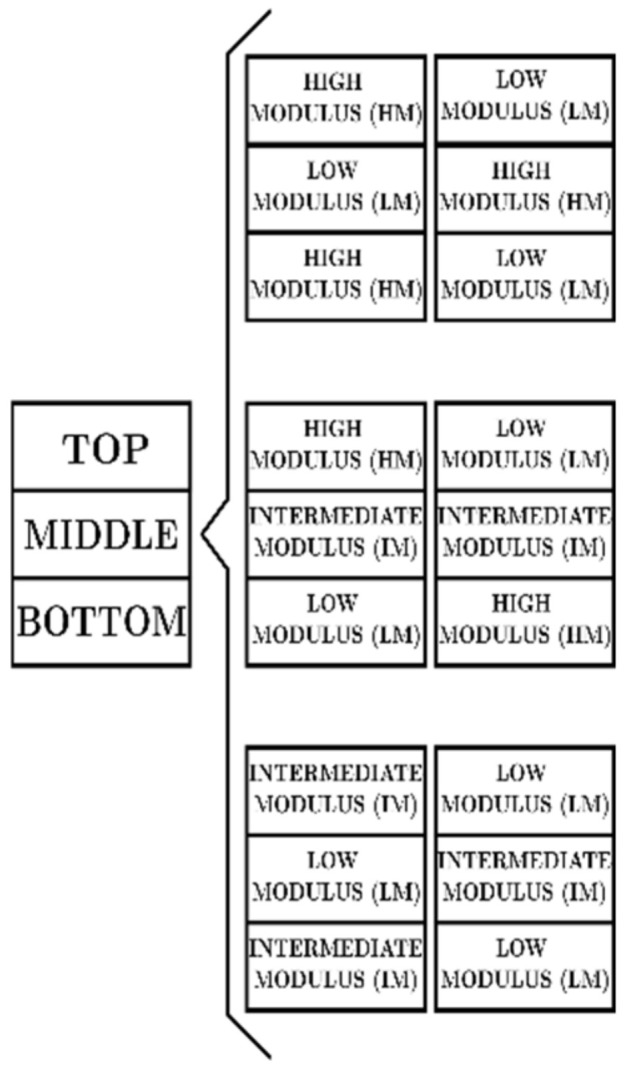
Layered lattice structures (LLS).

**Figure 6 materials-16-00649-f006:**
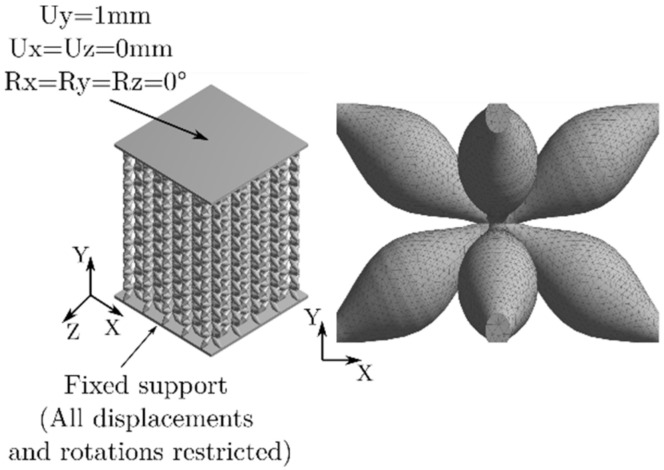
Left: Boundary conditions for FEA simulation. Right: Unit cell showing mesh size used.

**Figure 7 materials-16-00649-f007:**
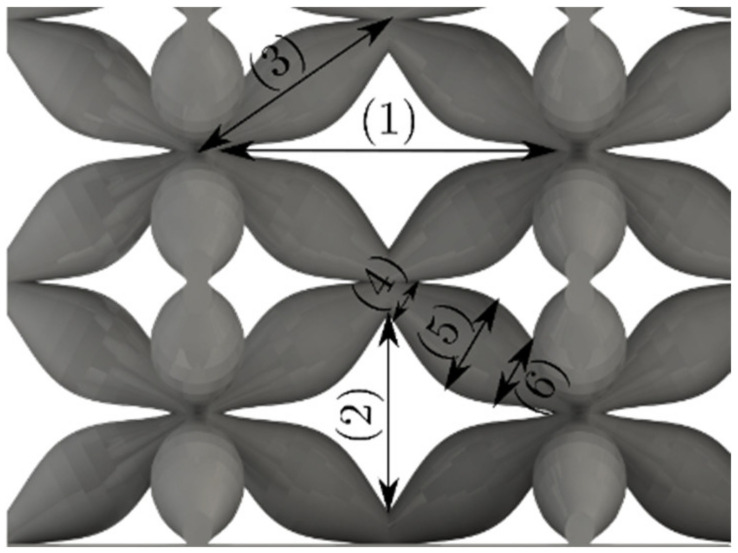
Six measurements were used to track variation between CAD and 3D printed struts.

**Figure 8 materials-16-00649-f008:**
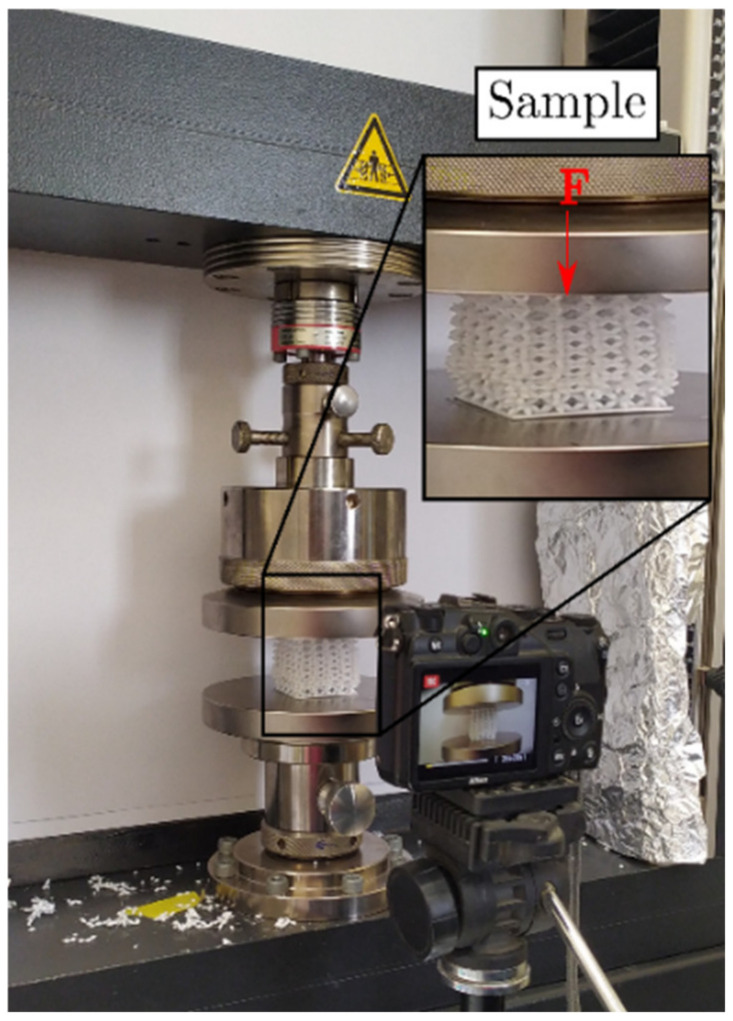
Experimental setup for compressive tests.

**Figure 9 materials-16-00649-f009:**
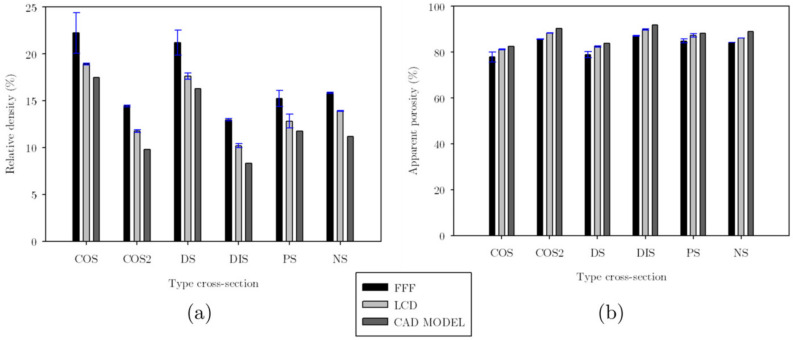
Comparison of the actual relative density (**a**) and the apparent porosity (**b**) of the additively manufactured samples and the computational models for UPLS.

**Figure 10 materials-16-00649-f010:**
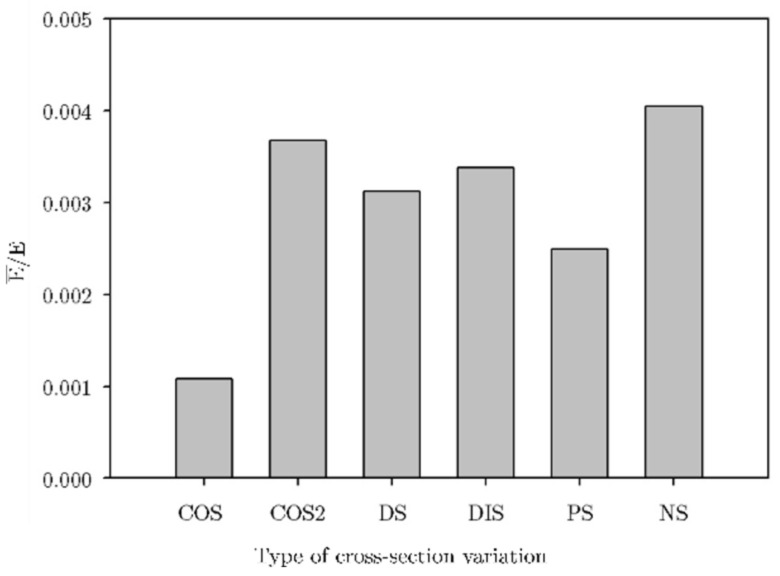
Normalized apparent Young’s modulus of uniformly periodic lattice structures via simulation.

**Figure 11 materials-16-00649-f011:**
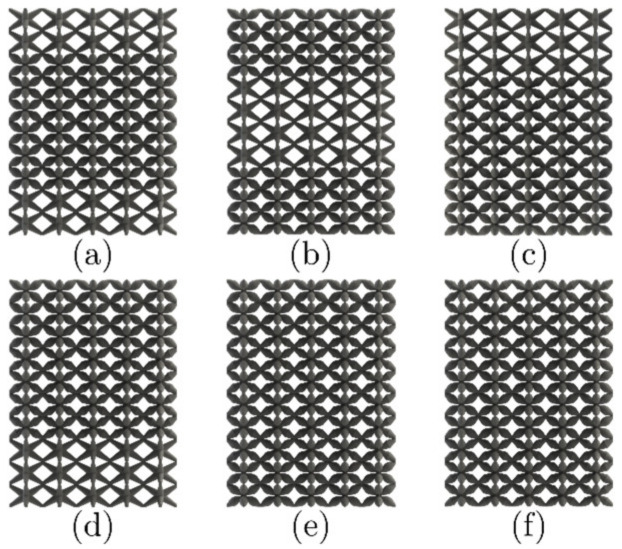
Layered lattice structures. (**a**) HM-LM-HM (**b**) LM-HM-LM (**c**) HM-IM-LM (**d**) LM-IM-HM (**e**) IM-LM-IM (**f**) LM-IM-LM.

**Figure 12 materials-16-00649-f012:**
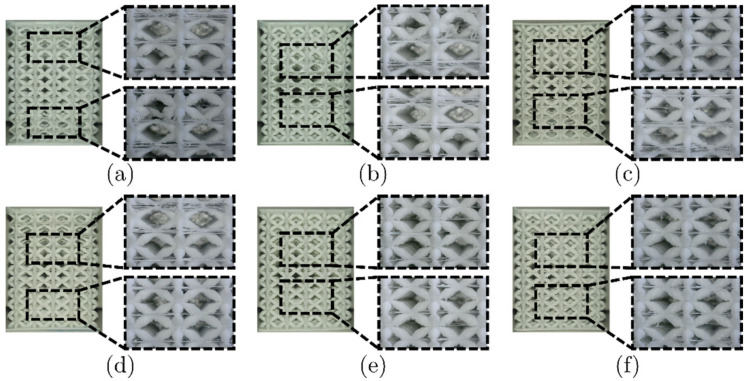
LLS printed via FFF: (**a**) HM-LM-HM, (**b**) LM-HM-LM, (**c**) HM-IM-LM, (**d**) LM-IM-HM, (**e**) LM-IM-LM and (**f**) IM-LM-IM.

**Figure 13 materials-16-00649-f013:**
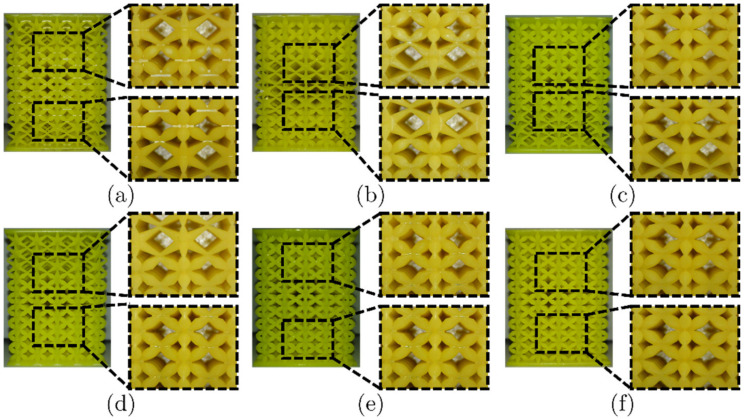
LLS printed via LCD: (**a**) HM-LM-HM, (**b**) LM-HM-LM, (**c**) HM-IM-LM, (**d**) LM-IM-HM, (**e**) LM-IM-LM and (**f**) IM-LM-IM.

**Figure 14 materials-16-00649-f014:**
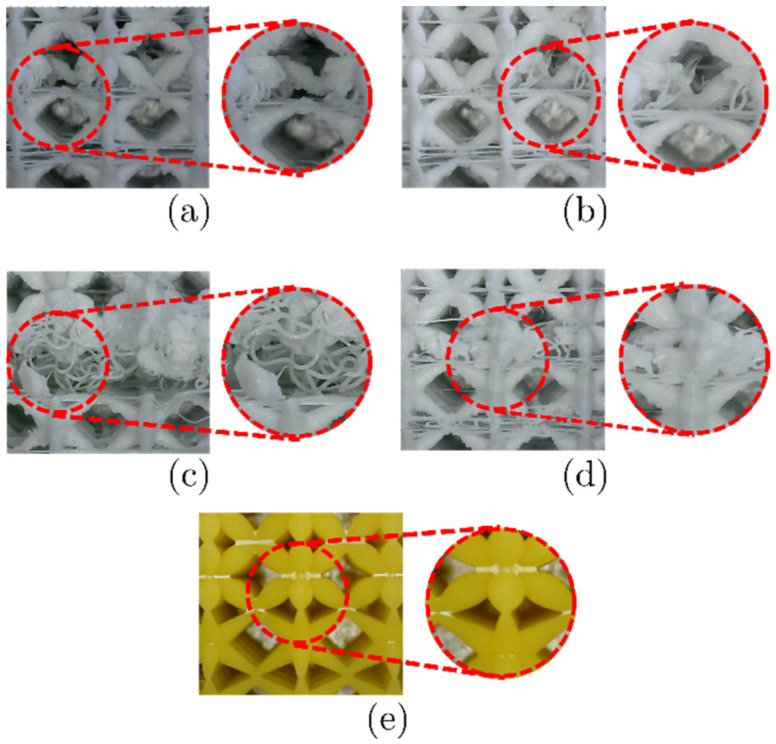
Print defects in LLS with PS and COS unit cells: (**a**) no connection in struts, (**b**) absence of struts, (**c**) absence of unit cells, (**d**) accumulation of material, and (**e**) connection in LCD structure.

**Figure 15 materials-16-00649-f015:**
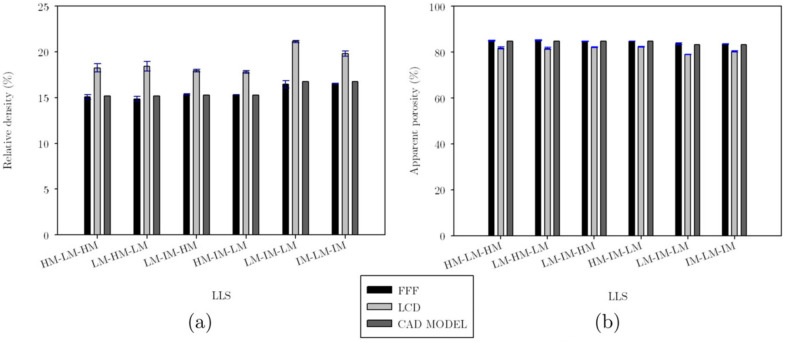
Comparison of the actual relative density (**a**) and the apparent porosity (**b**) of the additively manufactured samples and the computational models for LLS.

**Figure 16 materials-16-00649-f016:**
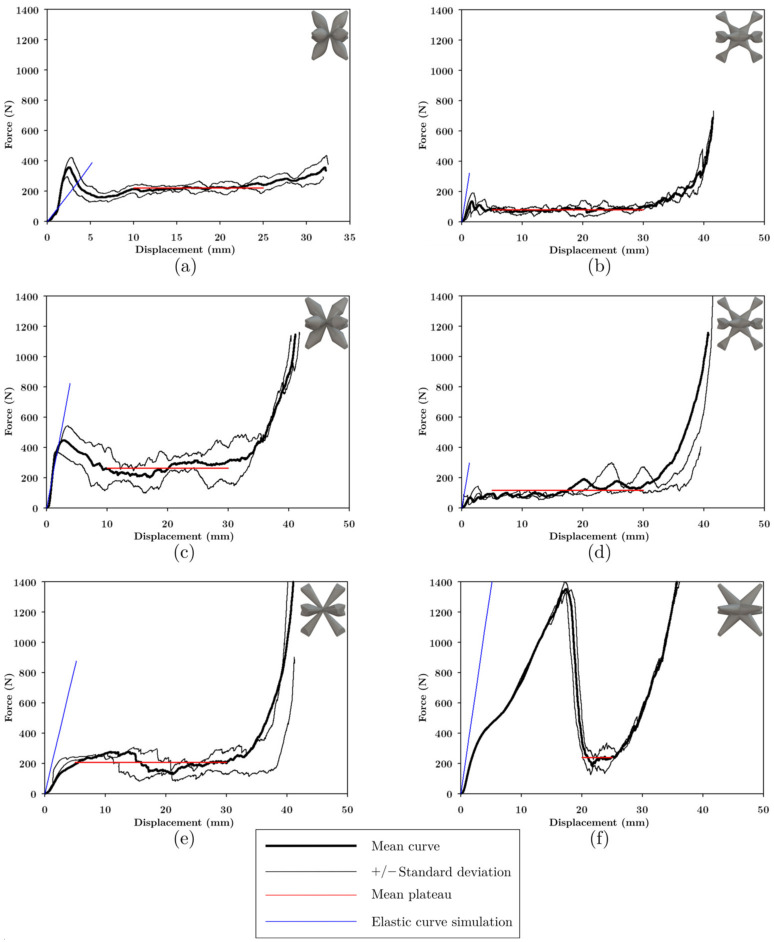
Force-Displacement curve of UPLS manufactured with PLA. For each variation in strut cross-section: (**a**) COS, (**b**) COS2, (**c**) DS, (**d**) DIS, (**e**) PS, and (**f**) NS.

**Figure 17 materials-16-00649-f017:**
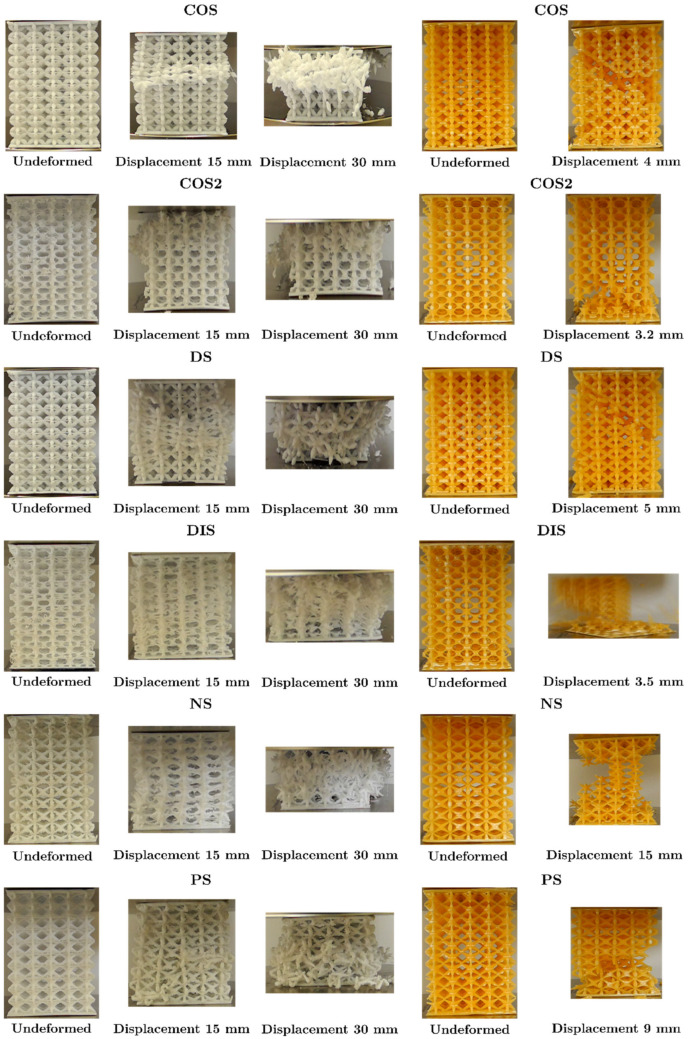
Deformation stages during the compression test. Left: lattice structures fabricated with PLA; Right: lattice structures fabricated with resin.

**Figure 18 materials-16-00649-f018:**
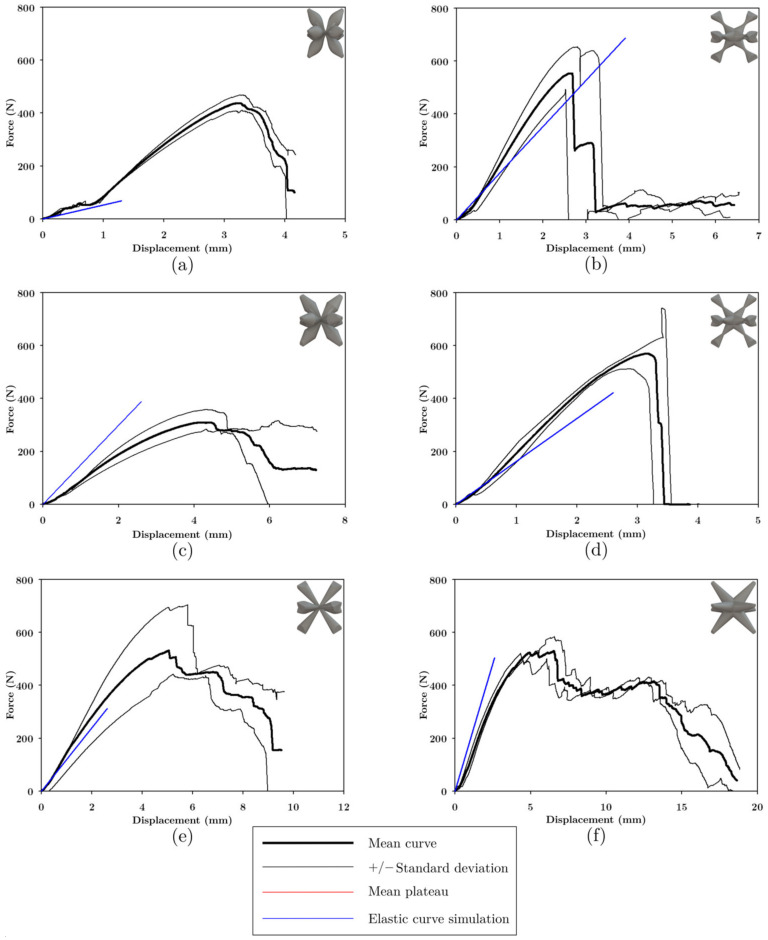
Force-displacement curve of UPLS manufactured with resin. For each variation in strut cross-section (**a**) COS, (**b**) COS2, (**c**) DS, (**d**) DIS, (**e**) PS, and (**f**) NS.

**Figure 19 materials-16-00649-f019:**
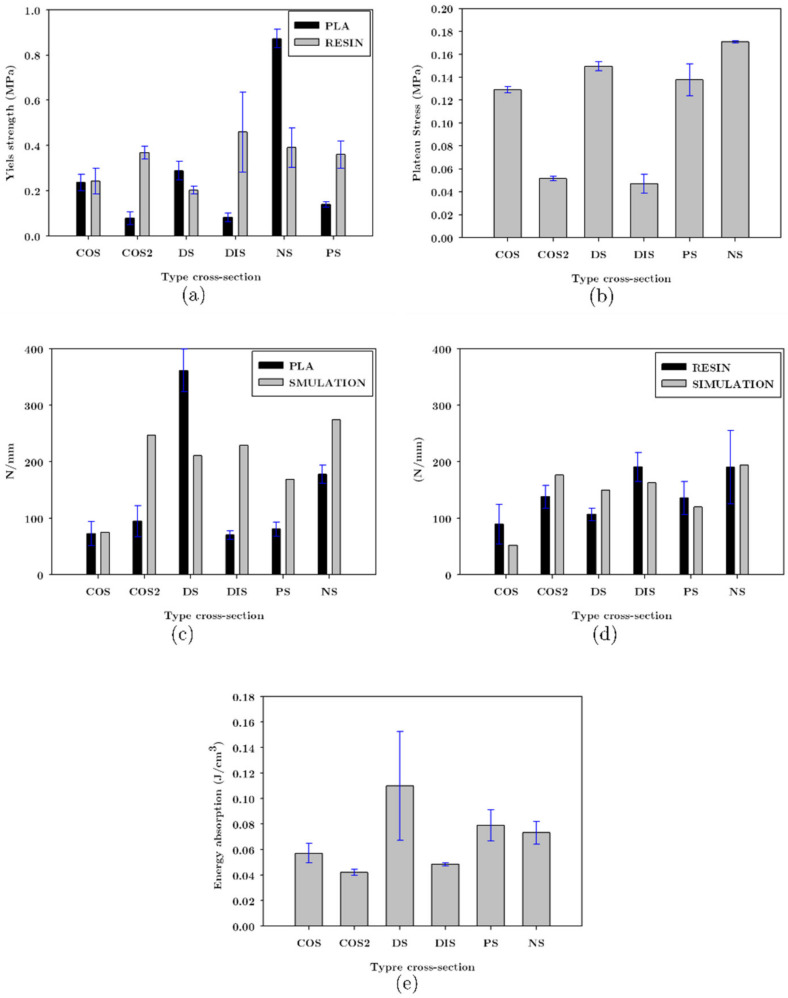
Comparative mechanical properties graph between lattices structures UPLS manufactured (**a**) Yield strength, (**b**) Plateau stress, (**c**) Stiffness comparative between experimental FFF and FEA, (**d**) Stiffness comparative between experimental LCD and FEA, and (**e**) Energy absorption of PLA lattice structures.

**Figure 20 materials-16-00649-f020:**
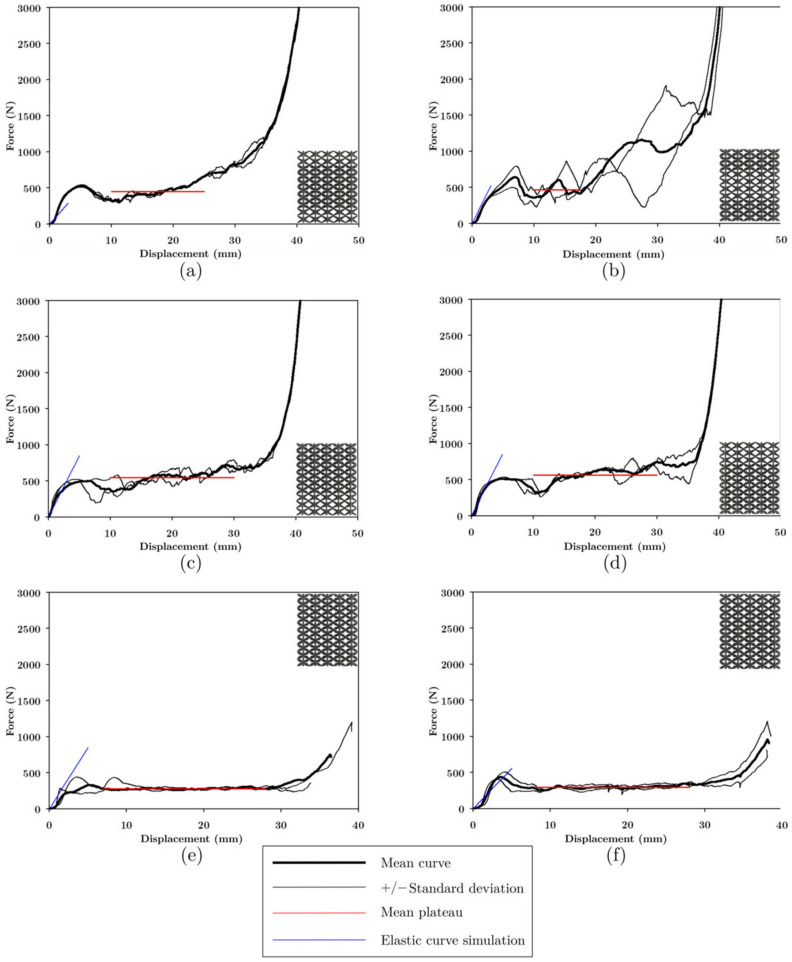
Force-displacement curve of LLS manufactured with PLA. (**a**) HM-LM-HM, (**b**) LM-HM-LM, (**c**) HM-IM-LM, (**d**) LM-IM-HM, (**e**) LM-IM-LM, and (**f**) IM-LM-IM.

**Figure 21 materials-16-00649-f021:**
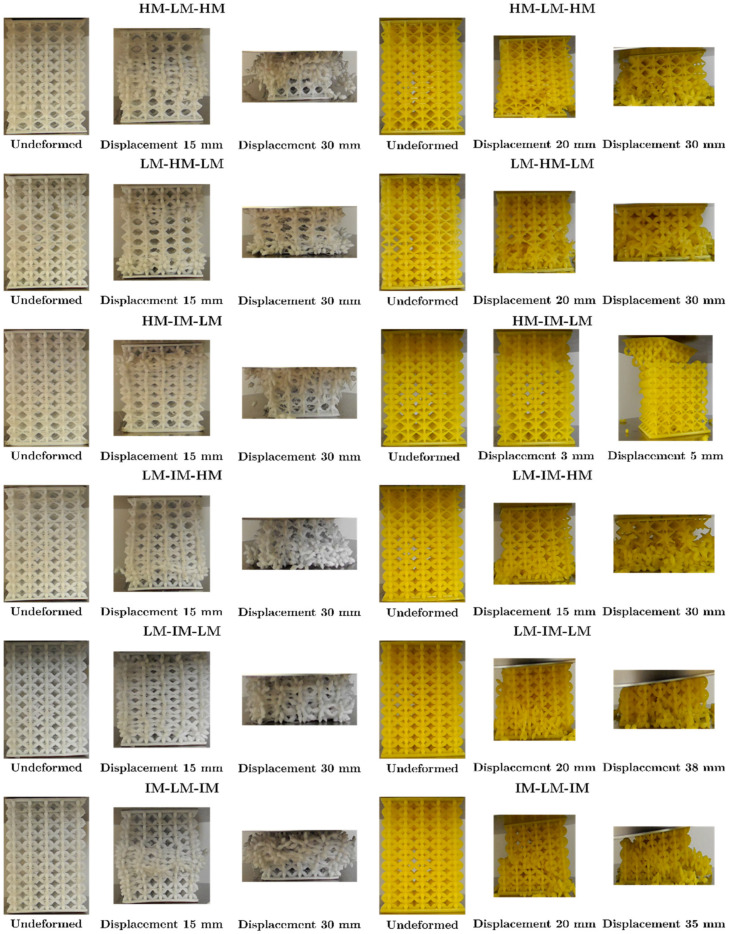
Deformed shape in a compression test. Left: lattice structures with PLA Right: lattice structures with resin.

**Figure 22 materials-16-00649-f022:**
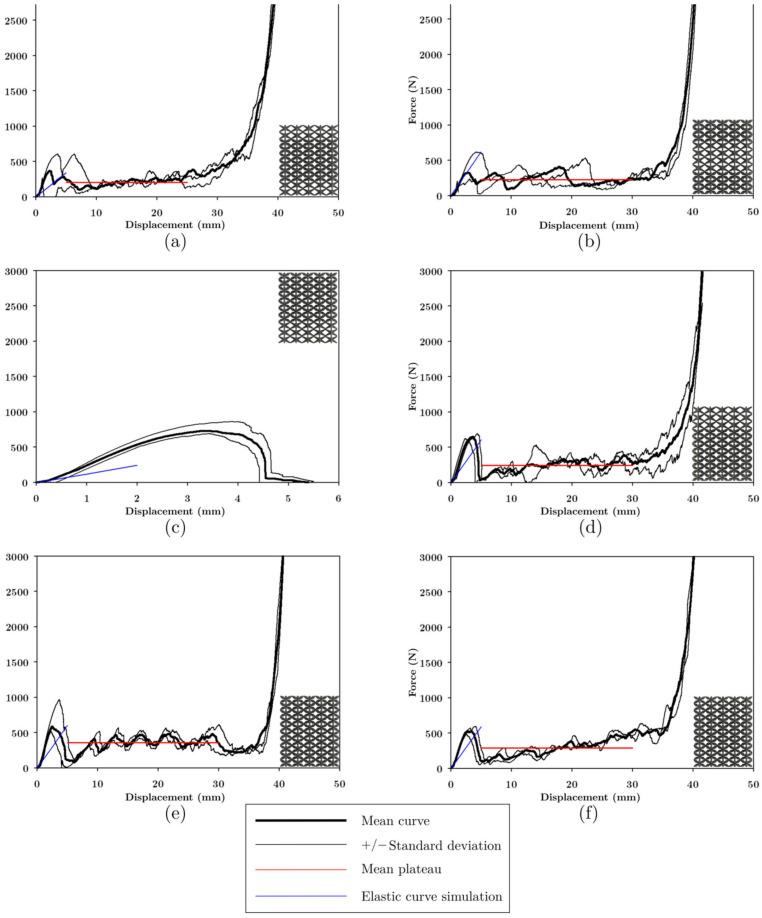
Force-displacement curve of LLS manufactured with resin: (**a**) HM-LM-HM, (**b**) LM-HM-LM, (**c**) HM-IM-LM, (**d**) LM-IM-HM, (**e**) LM-IM-LM, and (**f**) IM-LM-IM.

**Figure 23 materials-16-00649-f023:**
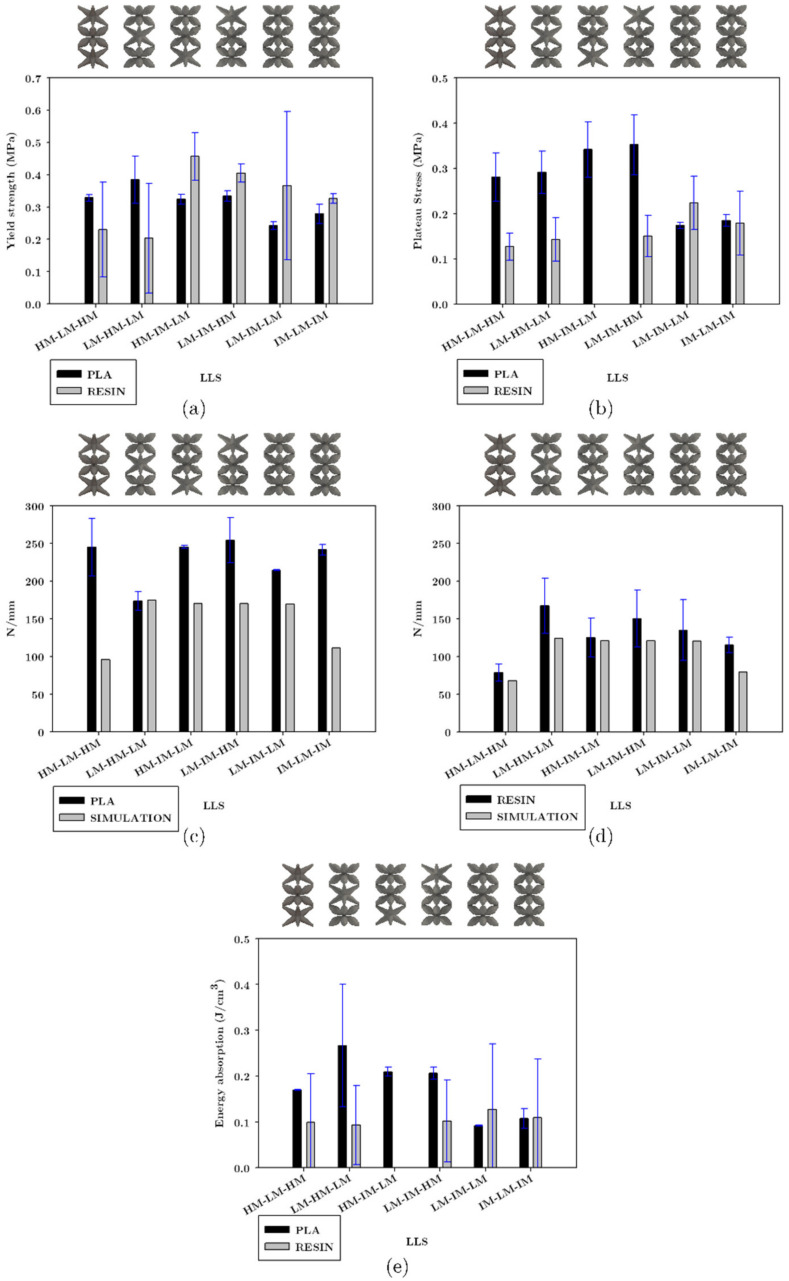
Comparative mechanical properties graph between lattices structures LLS manufactured: (**a**) yield strength, (**b**) plateau stress, (**c**) stiffness comparative between experimental FFF and FEA, (**d**) stiffness comparative between experimental LCD and FEA, and (**e**) energy absorption of PLA and resin lattice structures. Representative distribution of layers is included on top of each bar graph for the reader’s convenience.

**Table 1 materials-16-00649-t001:** Identification of the dimensions of the 3D printed models with the computational ones.

Cross-Section	3D Model	FFF	LCD
COS	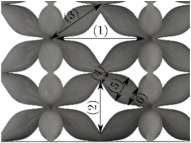	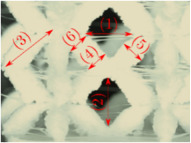	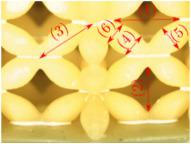
COS2	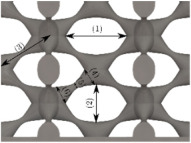	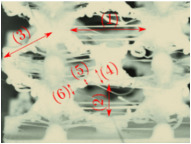	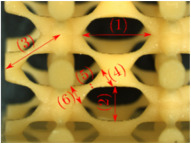
DS	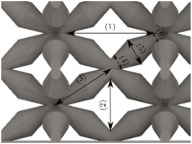	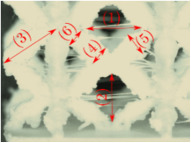	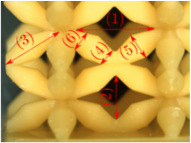
DIS	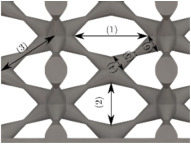	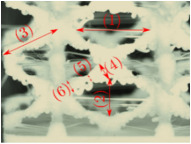	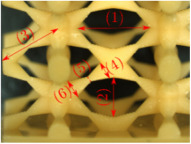
PS	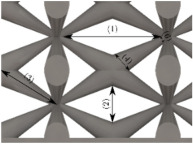	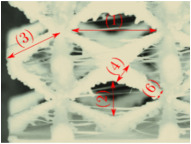	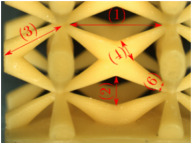
NS	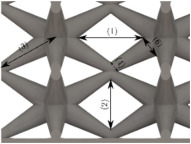	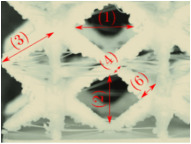	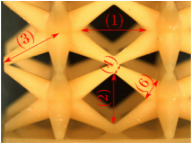

**Table 2 materials-16-00649-t002:** Measurement of parameters in computational models and 3D printed models.

FFF							
Cross-Section	CAD Model and Printing Process	Horizontal Space (1)	Vertical Space (2)	Strut Length (3)	Extreme Diameter (4)	Central Diameter (5)	Extreme Diameter (6)
COS	CAD	7.08	3.75	4.72	0.93	1.98	1.44
	Printed Average	3.18	3.5	3.475	1.345	1.68	1.515
COS2	CAD	5.12	2.76	4.72	1.44	0.49	1.44
	Printed Average	5.1	2.385	3.895	1.032	0.554	0.944
DS	CAD	7.2	3.75	4.72	1.35	2	1.12
	Printed Average	3.89	3.345	4.455	1.41	1.715	1.19
DIS	CAD	5.39	3.01	7.2	1.12	0.5	1.3
	Printed Average	5.34	2.53	4.22	0.965	0.49	0.93
PS	CAD	4.84	3.89	4.72	0.63	-	1.62
	Printed Average	4.69	3.405	4.225	0.6825	-	1.21
NS	CAD	6.91	2.69	4.72	1.67	-	0.59
	Printed Average	6.145	2.385	4.325	1.38	-	0.477
LCD							
Cross-section	CAD model and printing process	Horizontal Space (1)	Vertical Space (2)	Strut Length (3)	Extreme Diameter (4)	Central Diameter (5)	Extreme Diameter (6)
COS	CAD	7.08	3.75	4.72	0.93	1.98	1.44
	Printed Average	4.62	3.085	4.34	1.685	1.925	1.715
COS2	CAD	5.12	2.76	4.72	1.44	0.49	1.44
	Printed Average	4.81	2.54	4.3	1.26	0.483	1.455
DS	CAD	7.2	3.75	4.72	1.35	2	1.12
	Printed Average	5.55	3.095	4.535	1.295	1.875	0.965
DIS	CAD	5.39	3.01	7.2	1.12	0.5	1.3
	Printed Average	5.035	2.755	4.485	1.255	0.4	1.316
PS	CAD	4.84	3.89	4.72	0.63	-	1.62
	Printed Average	4.49	3.56	4.405	0.708	-	1.62
NS	CAD	6.91	2.69	4.72	1.67	-	0.59
	Printed Average	6.505	2.225	4.57	1.58	-	0.596

**Table 3 materials-16-00649-t003:** Data obtained from compression tests for UPLS structures.

UPLS Manufactured via the FFF Process			
Cross-Section	Maximum Load (N)	Displacement where the Maximum Load Occurs (mm)	Densification Load (N)
COS	357.8	2.5	280
COS2	134.8	1.37	101.9
DS	447.6	2.8	309.8
DIS	70.15	1.4	145.1
PS	276.3	10.9	247.6
NS	1351.7	17.4	243.9
UPLS manufactured via the LCD process			
Cross-section	Maximum load (N)	Displacement where the maximum load occurs (mm)	Densification load (N)
COS	437.3	3.2	-
COS2	553.2	2.6	-
DS	309.6	4.1	-
DIS	567.2	3.2	-
PS	531.4	5	-
NS	523.7	4.9	-

**Table 4 materials-16-00649-t004:** Data obtained from compression tests for LLS structures.

LLS Manufactured via the FFF Process			
Layered Lattice Structure	Maximum Load (N)	Displacement where the Maximum Load Occurs (mm)	Densification Load (N)
HM-LM-HM	524.6	5.25	815.2
LM-HM-LM	642.4	6.93	1095.2
HM-IM-LM	501.6	6.6	707
LM-IM-HM	513.5	5.14	755.96
LM-IM-LM	330.2	5.07	319
IM-LM-IM	437.2	3.56	340.7
LLS manufactured via the LCD process			
Layered Lattice Structure	Maximum Load (N)	Displacement where the Maximum Load Occurs (mm)	Densification Load (N)
HM-LM-HM	364.03	2.36	399.3
LM-HM-LM	325.4	2.9	246.4
HM-IM-LM	731.06	3.33	-
LM-IM-HM	648.1	3.59	307.4
LM-IM-LM	589.4	2.47	297.6
IM-LM-IM	521.5	3.48	476.8

## Data Availability

All data generated and analyzed in this article are included within the article.

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
