# Peer review of "Additively Manufactured Lattice Materials with a Double Level of Gradation: A Comparison of Their Compressive Properties when Fabricated with Material Extrusion and Vat Photopolymerization Processes"

_materials, 2023, doi:10.3390/ma16020649_

Round 1

Reviewer 1 Report

The article presents a comparative study on the performance of lattice-based structures made of unit cells with differing strut geometries. The authors designed and fabricated lattices with these unit cell geometries and also with uniformly periodic and layered architectures. The authors used both material extrusion and vat photopolymerization additive manufacturing processes. I believe the work done here is interesting but needs to be organized better with a clear identification of what new contributions are being made.

Here are concerns regarding this paper:

  • While the article has some very interesting ideas, it does not clearly state the importance of the current work in the context of all the previous work done on lattice structures. The Introduction section just lists some work from the literature without providing any context and relation to the current work.
  • It is not clearly motivated why the specific strut variations (COS, DS etc.) studied here are chosen. For example, why not use a Gaussian cross-section variation?
  • The term layered was used without properly defining it in the Introduction section although it becomes apparent in the subsequent sections.
  • A detailed discussion on how the unit cell dimensions were arrived at based on the printer resolutions is highly recommended.
  • It is clear that the FFF printer's resolution is inferior to that of the LCD-based printer. Furthermore,  material extrusion inherently introduces some anisotropy and the materials used were different. Consequently, it is expected that the performance of structures made by these processes will be different and so, it is not clear why it warrants a detailed comparison at all. The article perhaps could do without the results obtained with the FFF specimen.
  • The remarks made by the authors regarding the FFF specimen (poor adhesion, under/over extrusion) suggests that the FFF printer was not as well tuned as the LCD printer.
  • It is not clear what purpose the simulations serve if the dimensions/geometries of the experimental specimen and the simulation model are not the same. The simulation model could have been modified to reflect the actual printed dimensions.
  • It would have been a more interesting exercise had the relative density of the different unit cells are the same. In this case, the comparison will have revealed the true influence of the geometry on the mechanical properties.
  • Why are the 6 specific permutations for layered lattices used? For example, why not IM-IM-HM? This needs to be explained.
  • The results section can be better organized, perhaps in a tabular format, without having to just stating the values from the plots.
  • A few grammatical/typing/other concerns:
    • Why is VT used to abbreviate vat photopolymerization? Should it not be VP?
    • The sentence on line 27 "Differences in material supplier nature exhibit…" is not clear and could be re-phrased.
    • TPMS and honeycombs are periodic lattices too, why are they being grouped separately?
    • The phrase "adjust the mechanical properties" could be changed to "modify/tailor the mechanical…"
    • On line 199, it should be "…parameters were kept constant for all…"
    • On line 237, it should "…simulations and experiments."
    • On lines 678 and 684, it should be gradation instead of graduation.

Author Response

Author’s response to reviewer’s comments

ID: Material-2070597

Title: Additively manufactured lattice materials with a double level of gradation: a comparison of their compressive properties when fabricated with material extrusion and vat photopolymerization processes

We thank reviewer 1 for their time reading our manuscript, their useful comment, and suggestions for improvements. Below are our itemized responses, outlining changes made during the revision of the manuscript. These changes are also colored in yellow in the manuscript to facilitate version comparison. The following color scheme has been used: (i) Reviewer comment in blue and (ii) author´s response in Black Bolded font.

The article presents a comparative study on the performance of lattice-based structures made of unit cells with differing strut geometries. The authors designed and fabricated lattices with these unit cell geometries and also with uniformly periodic and layered architectures. The authors used both material extrusion and vat photopolymerization additive manufacturing processes. I believe the work done here is interesting but needs to be organized better with a clear identification of what new contributions are being made.

We are very thankful for the appreciation of our work and pleased to know that you found our work interesting. Below we respond to each of this reviewer’s suggestions and comments.

Here are concerns regarding this paper:

  1. While the article has some very interesting ideas, it does not clearly state the importance of the current work in the context of all the previous work done on lattice structures. The Introduction section just lists some work from the literature without providing any context and relation to the current work.

We appreciate the interest shown in our work. To highlight the relationship between the reviewed results in the literature and ours, in addition to establishing the contribution of our work, we have added the following lines at the end of these paragraphs in the literature review: 

Sentence added in a paragraph, line 57 of the revised document.

Liu et al.[15] created functionally graded porous structures using gyroid and diamond unit cells. The three gradients proposed in [15] were achieved by varying the density, heterostructure, and cell size, achieving comparable mechanical properties to cortical bone. Bai et al. [9] studied additively manufactured FGLS built using selective laser sintering (SLS). The topologies studied were body-centered cubic (BCC) by varying the unit cell size and the diameter of the strut in a unidirectional way. Then, the compression properties were characterized through experimental tests and numerical simulations, with errors of  %,  %, and %, for stiffness, strength, and plateau stress, respectively. Zhang et al. [16] analyzed structures built with a combination of unit cells (simple cubic and octet) via SLS, including the defects at the interfaces between topologies. The topology variation in the structure allowed it to control its strength, stiffness, and energy absorption. These structures presented instability in their deformation due to the connection interfaces of the gradients and resulted in being more flexible than periodical structures. Dumas et al. [17] developed a graded cell structure applying a scale factor in the nodes of the diamond unit cell and studied its compression response both numerically and experimentally. When comparing the experimental results with the simulation results, they present a divergence of % for stiffness and % for yield strength. Li et al. [18] studied (theoretically and experimentally) the compression properties of lattice sandwich structures with variable cross-sections. The structures were manufactured using stereolithography (SLA), and the strength of the structures with variable cross-sections was greater than structures with uniform cross-sections. These works have covered the characterization of the mechanical properties in structures with variations of the topology, unit cell size, or relative density; however, they do not explore the feature of varying the geometry of the strut cross-section, which could be an additional parameter to study towards the tailoring of the effective properties of porous media.

Sentence added in a paragraph, line 89 of the revised document.

FGLS have been explored to mimic the bone structure, density, and stiffness of trabecular or cancellous bone; often needed in prosthesis design to allow bone growth and to fix the implant naturally with human bone [20,21]. Wang et al. [22] used a graded octet cell structure to design an acetabular component to improve implant-bone stability; the graded cell structure allowed the elastic modulus of the implant to be matched with that of bone. Alkhatib et al. [23] studied the load transfer towards the femur from the stem of the hip prosthesis built with periodic and graded structures using BCC unit cells, discovering that stems with graduated structures transfer a more significant load to the femur than stems with structures uniformly periodic. The implementation of lattice structures to design implants as hip prosthesis could be a potential application of lattice structures proposed in this work, as here, an additional level of parameter variation is presented; aiming to achieve a wider range of mechanical properties.

  1. It is not clearly motivated why the specific strut variations (COS, DS, etc.) studied here are chosen. For example, why not use a Gaussian cross-section variation?

We thank the reviewer for this observation; we agree that a justification of the selection of struts cross-section variation is missing. In this work, the main purpose was to vary in a controlled (mathematically parametrized) manner such as struts cross-section. The intention was to achieve two types of variations, one in which the strut gets thicker at the midpoint of its length, and another where it gets thicker at the opposite sides. We achieved this by two mathematical approaches: (i) cosine function and (ii) linear function. The former allows us a smoother transition in the cross-sections. As this reviewer accurately pointed out, this explanation is missing from the manuscript; hence we have added a sentence in the second paragraph of section 2.1 (lines 151 to 153).

Of course, other mathematical parametrizations to achieve these variations could be explored, e.g. Gaussian. However, we believe that with the proposed in this paper, the objective was properly achieved.

Sentence added in a paragraph, line 147 of the revised document.

The first level, with the aim of tailoring the mechanical properties of lattice structures, is to grade varying strut cross-sections along their length. Six cross-section variations were considered: cosine (COS), cosine 2 (COS2), double slope (DS), double inverted slope (DIS), positive slope (PS), and negative slope (NS). This mathematical parameterization is proposed to vary the diameter of the struts in the unit cell achieving control of the accumulation of material in the center and at the ends of the strut’s length. These variations are plotted in Figure 3, along with the resulting unit cells. The corresponding lattice structures formed with these unit cells are then depicted in Figure 4 ( unit cells). The relative density  of the lattice structures was calculated using the volume fraction as: , where  is the volume of the actual space occupied by the lattice structure, and  is the overall volume of the lattice structure, including the empty spaces.

  1. The term layered was used without properly defining it in the Introduction section although it becomes apparent in the subsequent sections.

We apologize for this inattention. To define the term layered lattice, we have added the following line to the second paragraph of the introduction:

Sentence added in a paragraph, line 45 of the revised document.

Lattice structures are built from unit cells uniformly distributed within a given volume, known as periodic lattice structures. On the other hand, some structures can also be found with gradual variations of topology [7] or parameters of the unit cell [8]; these are known as functionally graded lattice structures (FGLS) [9]. Other types of topology variation are known as layered lattice structures (LLS); these are structures where variations are defined at different layers of unit cells [10].

  1. A detailed discussion on how the unit cell dimensions were arrived at based on the printer resolutions is highly recommended.

To determine the printing parameters in the FFF process, tests were carried out modifying two parameters, nozzle size, and printing speed. The printing of the cell structure with unit cells of the size mentioned in the manuscript was achieved with a 0.2mm nozzle and a speed of 40mm/s. Because the diameter in the narrowest area of the prop measures 0.4mm, it was not feasible to use a larger nozzle, also, if a higher speed than the mentioned one was considered, the struts would break at the moment of extrusion of the material. Although it was possible to reduce the speed and the size of the nozzle to improve the formation of the struts, it was sought to have a balance between the quality and the printing time due to the number of samples to be printed.

To detail how the parameters were selected for printing the structures in the FFF process, the paragraphs marked in sections 2.3 and 3.1 were added.

Lines added in a paragraph, line 201 of the revised document.

The lattice structures were additively manufactured using two printing technologies: FFF and LCD; a Creality® Ender 3D printer was used for FFF, and a Creality® LD002-R printer for LCD. Computational models (STL files) for the FFF method were processed in Ultimaker Cura® V4.8.0 (Ultimaker B.V., Utretch, The Netherlands) software, and PLA material supplier (in the form of filament) was used as raw material. To determine the manufacturing parameters for the FFF process, tests were carried out with different sizes of unit cells, nozzles, and printing speed; in addition, a displacement in the Z-direction of 0.1mm was set to the printing head, when it moved from one point to another. Finally, the minimum size of the unit cell that could be printed was shown in section 2.1 using the following parameters: extruder nose diameter of mm, layer height of mm, infill, bed temperature of °C, extruder temperature of °C and printing speed mm/s. On the other hand, models (STL files) for LCD were processed using Chitubox® V1.6.2 software; samples were made with Creality® photosensitive resin. The layer thickness was mm; the exposure time was s and the exposure time for the first layer with s. Isopropyl alcohol was used to remove the excess liquid resin from the samples. The samples were then cured by exposing them to ultraviolet light for min.

In both manufacturing processes, the printing parameters were not optimized, i.e., standard parameters were used. The printing parameters were kept constant for all samples and printed without supports, and no special methods were required for adhesion to the print bed. All samples were fabricated-oriented, so the XZ-plane was parallel to the printing platform. The weight of the samples was measured with a RADWAG AS 220.RS balance to subsequently calculate the relative density.

Lines added in a paragraph, line 256 of the revised document.

The FFF fabrication took up to h for each structure, while the simultaneous fabrication of two structures took up to h with LCD. The main parameters that allowed the fabrication of the structures in FFF were the diameter of the nozzle of 0.2 mm and the printing speed of 40 mm/s. Because the minimum diameter of the struts at their thinnest area was 0.4mm, printing with a larger diameter nozzle was not possible. The maximum speed that could be applied was 40mm/s because if this parameter was increased, the struts of the COS2 and DIS structures would bent, due to being thin in the center of their length. On the other hand, the speed did not decrease further speed because the printing time increased considerably.

  1. It is clear that the FFF printer's resolution is inferior to that of the LCD-based printer. Furthermore,  material extrusion inherently introduces some anisotropy and the materials used were different. Consequently, it is expected that the performance of structures made by these processes will be different and so, it is not clear why it warrants a detailed comparison at all. The article perhaps could do without the results obtained with the FFF specimen.

We agree with you in your observations regarding the FFF process; however, in this work, we wanted to evaluate the feasibility of using both printing processes for the manufacture of structures with diameter variation in the struts and distribution by layer. The comparison of printing defects in both manufacturing processes was discussed qualitatively, e.g., it is mentioned that in the FFF process, there is no adhesion of material in the narrow areas of the prop; in the nodes, there is an accumulation of material, and detachment is appreciated of drop-shaped material on the periphery of the strut, also in the transition interfaces in the LLS structures printed with this process there was no connection between the struts. On the other hand, in the LCD printing process, these defects did not appear, and the struts had a greater geometric similarity with the CAD model. Finally, direct comparisons are presented to give the reader an insight as to what to expect in each AM technique, not to diminish one or the other. The inclusion of results of these two types of technologies is frequently reported in the literature (Guerra Silva et al. 2021). We believe that including the mechanical and manufacturing results of both technologies could serve as a decision framework for the interested community. If the reviewer still feels the FFF results should be removed, we can easily do so in a revised version of the manuscript.

Guerra Silva, R., Torres, M. J., & Zahr Viñuela, J. (2021). A Comparison of miniature lattice structures produced by material extrusion and vat photopolymerization additive manufacturing. Polymers, 13(13), 2163.

  1. The remarks made by the authors regarding the FFF specimen (poor adhesion, under/over extrusion) suggest that the FFF printer was not as well tuned as the LCD printer.

Thank you for this comment. We agree that it appears that the FFF printer was not tuned; we would like to clarify that none were tuned. The use of both printers was done with the default manufacturing parameters. The only aspect considered regarding the fabrication with FFF was the design of lattices with struts inclined at an angle that did not need support. No work was done on the optimization of the manufacturing parameters in any of the processes because the main objective of the investigation is not to characterize manufacturing defects; we did present the defects as we believe they add important information to the reader as to what to expect. Also, the characterization of printing defects was considered one reason for the discrepancy in the results between simulation and measurements. We agree that the optimization of the FFF printing parameters would lead to better fabrication results. However, a whole optimization study is needed for these could easily lead to another independent story to tell. For sure, this study could be a future work complementary to this research.

  1. It is not clear what purpose the simulations serve if the dimensions/geometries of the experimental specimen and the simulation model are not the same. The simulation model could have been modified to reflect the actual printed dimensions.

We appreciate this comment. The study of the compression properties of the cellular structures was carried out in simulation to corroborate the results obtained experimentally. While the same trend was observed computationally and experimentally, some differences also resulted. The resulting discrepancies between the simulation and experimental results led us to explore the reasons further. As these were attributed to the defects obtained, these were further characterized using a micrograph. We agree that a refined FE model that considers the defects would be of great value; this is not trivial. The technique used for the measurement of the unit cells only allows observing the defects in one plane; therefore, the shapes of the strut are not fully observed nor characterized. Assuming that the deviations from the CAD model are the same in every direction could result in an inaccurate FE model representation.

  1. It would have been a more interesting exercise had the relative density of the different unit cells are the same. In this case, the comparison will have revealed the true influence of the geometry on the mechanical properties.

We agree that the property-relative density relation is of great value in the lattice and cellular materials community. The purpose here was not to match the relative densities because the objective was to have the same parametric definition of the strut cross-section variation. When achieving this, the relative densities varied, as the accumulation of material changed depending on the strut cross-section variation. If attempting to match the relative densities, the parameters used in the mathematical definition of the cross-section will have to be altered to achieve this. Nevertheless, we do have similar relative densities comparison, as we can see, for instance, in Figure 9 (included here for the sake of reviewer convenience), COS and DS have the same relative density, and COS2 and DIS, also have similar relative densities even when compared with PS and NS.

Figure 9. Comparison of the actual relative density of the additively manufactured samples and the computational models.

The aspect of matching relative densities was also covered in the study of the Layered lattice structures. All these were designed to have similar relative densities. The maximum relative density difference among the LLS is 1.56%. Figure 15 and the following text were included in section 3.3 of the manuscript to highlight this.

Lines added in a paragraph, line 353 of the revised document.

Figure 15 shows the comparison of the relative density of the CAD models with the structures printed with both manufacturing processes. It can be seen that the CAD model does not present significant differences with the structures manufactured via FFF; this is mainly due to the absence of material in the unit cell type change interfaces because the structures manufactured via LCD had a perfect union in the interfaces and that in general, the unit cell struts were thicker than those of the CAD model, it is observed a greater difference when making the comparison. It should be noted that the computational models show a maximum difference of 1.56% of the relative density between the proposed permutations.

Figure 15. Comparison of the actual relative density of the additively manufactured samples and the computational models for LLS

  1. Why are the 6 specific permutations for layered lattices used? For example, why not IM-IM-HM? This needs to be explained.

We apologize for the missing justification as to why the permutations for the layered lattices were proposed. The objective was to propose layered arrangements such that there are no consecutive layers with the same unit cell strut cross-section variation. For instance, IM-IM-HM will have two consecutive layers with the same stiffness. While this is interesting to test in future works, these types of arrangements would not expose the influence of the transition between types of struts cross-sections. We agree with the reviewer that this reason is missing in the text, hence we have added these lines in section 2.1

Sentence added in a paragraph, line 167 of the revised document.

The second level controls the unit cell distributions layer-by-layer, i.e., layered lattice structures (LLS). Six LLS were designed considering an initial Finite Element simulation of the structures under compressive loading shown in Figure 4. From these simulations, their stiffness was measured so that these were labeled as lower elastic modulus (LM), intermediate elastic modulus (IM), and higher elastic modulus (HM). The lattice samples were divided into three zones (bottom, middle and top) to accommodate the unit cells with different elastic moduli, as illustrated in Figure 5. The number of unit cell layers in the zones mentioned above is adjusted for each proposed structure to match their relative densities. Figure 5 shows a labeling scheme used for the LLS arrangements studied: HM-LM-HM stands for a structure with a low elastic modulus in the middle, and a high elastic modulus in the top and bottom zones. The remaining structures studied have the following arrangements: LM-HM-LM, HM-IM-LM, LM-IM-HM, LM-IM-LM, and IM-LM-IM. The arrangements were proposed to obtain structures with three zones of different stiffness and also to avoid having the same stiffness in consecutive zones.

  1. The results section can be better organized, perhaps in a tabular format, without having to just stating the values from the plots.

We apologize for the dissatisfactory organization, and we appreciate your suggestion. Hence, we summarized the results in two tables that have been included in sections 3.4.3 and 3.4.4. These tables are included here for the reviewer's convenience. We also adjusted the text of sections 3.4.1, 3.4.2, and 3.4.4 so as not to be redundant with the data presented in the table.

Table added in section 3.4.1, line 411 of the revised document.

Table 3.  Data obtained from compression tests for the UPL structures

UPLS manufactured via the FFF process

Cross-section

Maximum load (N)

Displacement where the maximum load occurs (mm)

Densification load (N)

COS

357.8

2.5

280

COS2

134.8

1.37

101.9

DS

447.6

2.8

309.8

DIS

70.15

1.4

145.1

PS

276.3

10.9

247.6

NS

1351.7

17.4

243.9

UPLS manufactured via the LCD process

Cross-section

Maximum load (N)

Displacement where the maximum load occurs (mm)

Densification load (N)

COS

437.3

3.2

-

COS2

553.2

2.6

-

DS

309.6

4.1

-

DIS

567.2

3.2

-

PS

531.4

5

-

NS

523.7

4.9

-

Table added in section 3.4.4, line 619 of the revised document.

Table 4.  Data obtained from compression tests for LL structures

LLS manufactured via the FFF process

Layered lattice structure

Maximum load (N)

Displacement where the maximum load occurs (mm)

Densification load (N)

HM-LM-HM

524.6

5.25

815.2

LM-HM-LM

642.4

6.93

1095.2

HM-IM-LM

501.6

6.6

707

LM-IM-HM

513.5

5.14

755.96

LM-IM-LM

330.2

5.07

319

IM-LM-IM

437.2

3.56

340.7

LLS manufactured via the LCD process

Layered lattice structure

Maximum load (N)

Displacement where the maximum load occurs (mm)

Densification load (N)

HM-LM-HM

364.03

2.36

399.3

LM-HM-LM

325.4

2.9

246.4

HM-IM-LM

731.06

3.33

-

LM-IM-HM

648.1

3.59

307.4

LM-IM-LM

589.4

2.47

297.6

IM-LM-IM

521.5

3.48

476.8

  1. A few grammatical/typing/other concerns:

We apologize for the inattention in the grammatical errors and typos. We attended your observations, and they are listed below:

  • Why is VT used to abbreviate vat photopolymerization? Should it not be VP?

Line 22 to 23.

These were then additively manufactured via material extrusion (ME) and vat photopolymerization (VP).

  • The sentence on line 27 “Differences in material supplier nature exhibit…” is not clear and could be re-phrased

Line 27 to 28.

The brittle natural behavior of the resin caused a lack of plateau region in the stress-strain curves for the UPL structures, as opposed to those fabricated with ME.

  • TPMS and honeycombs are periodic lattices too, why are they being grouped separately?

Line 38 to 40.

They are often classified into 2D structures (honeycombs), and 3D structures. The latter includes random structures (foams),strut-based lattices, and triply periodic minimal surfaces (TPMS)  [3–5].

  • The phrase "adjust the mechanical properties" could be changed to "modify/tailor the mechanical…"

We have changed the writing in two phrases to:

Line 132.

This work aims to tailor the mechanical properties of lattice structures at two different levels: (i) functionally graded modification of the strut cross-section and (ii) layered distribution of unit cells.

Line 147.

The first level, with the aim of tailoring the mechanical properties of lattice structures, is to grade varying strut cross-sections along their length.

  • On line 199, it should be “… parameters were kept constant for all…”

Line 218.

The printing parameters were kept constant for all samples and printed without support.

  • On line 237, it should “… simulations and experiments”

Line 254

…and the comparison of stiffness determined by simulations and experiments.

  • On lines 678 and 684, it should be gradation instead of graduation

Line 691

The change of cross-section in the struts was the first level of gradation proposed, which allowed modifying the strength…

Line 697

The second level of gradation was implemented with the LLS, the experimental results…

Reviewer 2 Report

It is well known that the material and shape of reinforcement affect the strength characteristics of objects created using additive technologies.
Why did the authors choose such shapes of reinforcements for comparison?
Why were the PLA samples in fig. 17 evaluated with the same displacement values of 0, 15 and 30 mm, while for resin samples the evaluated displacement values differ significantly?
On the other hand, in fig. 21, the displacement values are similar for both types of material, but even here for the resin samples, the evaluated displacement values are not the same - for what reason?

Author Response

Author’s response to reviewer´s comments

ID: Material-2070597

Title: Additively manufactured lattice materials with a double level of gradation: a comparison of their compressive properties when fabricated with material extrusion and vat photopolymerization processes

We thank reviewer 2 for their time reading our manuscript, their useful comment, and suggestions for improvements. Below are our itemized responses, outlining changes made during the revision of the manuscript. These changes are also colored in yellow in the manuscript to facilitate version comparison. The following color scheme has been used: (i) Reviewer comment in blue and (ii) author´s response in Black Bolded font.

  1. It is well known that the material and shape of reinforcement affect the strength characteristics of objects created using additive technologies. Why did the authors choose such shapes of reinforcements for comparison?

We agree that the shape of the struts cross-section has a direct influence on the mechanical properties (both stiffness and strength) of the structure that is made from it. In this work, the main purpose was to vary in a controlled manner (mathematically parametrized), such as struts cross-section. The intention was to achieve two types of variations, one in which the strut gets thicker at the midpoint of its length and another where it gets thicker at the opposite sides. We achieve this by two mathematical approaches: (i) cosine function and (ii) linear function. The former allows us a smoother transition in the cross-sections. To clarify this for the reader, we have added a sentence in the second paragraph of section 2.1 as follows:

Sentence added in a paragraph, line 147 of the revised document.

The first level, with the aim of tailoring the mechanical properties of lattice structures, is to grade varying strut cross-sections along their length. Six cross-section variations were considered: cosine (COS), cosine 2 (COS2), double slope (DS), double inverted slope (DIS), positive slope (PS), and negative slope (NS). This mathematical parameterization is proposed to vary the diameter of the struts in the unit cell achieving control of the accumulation of material in the center and at the ends of the strut’s length. These variations are plotted in Figure 3, along with the resulting unit cells. The corresponding lattice structures formed with these unit cells are then depicted in Figure 4 ( unit cells). The relative density  of the lattice structures was calculated using the volume fraction as: , where  is the volume of the actual space occupied by the lattice structure, and  is the overall volume of the lattice structure, including the empty spaces.

  1. Why were the PLA samples in fig. 17 evaluated with the same displacement values of 0, 15, and 30 mm, while for resin samples the evaluated displacement values differ significantly?

Thank you for this comment; we agree that it needs further clarification. The differences in displacement values for each type of material are owed to their nature; PLA had a more ductile response, while resin samples had a brittle response, hence reaching a shorter displacement before failure. This was further explained in the manuscript by adding the following explanation to section 3.4.2, as shown below:

Sentence added in a paragraph, line 414 of the revised document.

Figure 17 shows the force-displacement graphs of the UPL structures manufactured via LCD. The structures manufactured with photosensitive resin did not present a plateau or densification; the maximum load of the structures occurred at a displacement of  mm to  mm; after the maximum load occurred, it fell abruptly due to the sudden detachment of the unit cells or fragments of the structure. This failure mode can be attributed mainly to the brittleness of the material. Figure 18 shows the deformation of the structures during the compression test. It was not possible to analyze the PLA and resin structures at the same range of displacement because the UPLS manufactured via LCD broke between 3mm and 6mm of displacement, while those of PLA rupture occurred at approximately 30mm of displacement.

  1. On the other hand, in fig. 21, the displacement values are similar for both types of material, but even here for the resin samples, the evaluated displacement values are not the same - for what reason?

Thank you for the observation, which is one of the more interesting findings of our work. As correctly noted in Figure 21, the displacements for both PLA and resin are closer in value, as opposed to what was observed in Fig. 17 (previous comment).  This is entirely due to the arrangement of layers. Layered samples changed their failure mechanism. This was observed as the different types of unit cell variations were used to induce the failure of different layers at different displacement values. For example, if a layer with a type of variation with low stiffness has failed, the others can still offer resistance to the load, somehow blocking the entire failure of the sample.

Round 2

Reviewer 1 Report

The text added between Figures 14 and 15 is not very clear and could be reworded. Otherwise, the authors have addressed all my concerns/comments.

Author Response

Author’s response to reviewer’s comments

ID: materials-2070597

Title: Additively manufactured lattice materials with a double level of gradation: a comparison of their compressive properties when fabricated with material extrusion and vat photopolymerization processes 

We thank reviewer 1 for their time reading our manuscript, their useful comment, and suggestions for improvements. Below are our itemized responses, outlining changes made during the revision of the manuscript. These changes are also colored in yellow in the manuscript to facilitate version comparison. The following color scheme has been used: (i) Reviewer comment in blue and (ii) author´s response in Black Bolded font.

The text added between Figures 14 and 15 is not very clear and could be reworded. Otherwise, the authors have addressed all my concerns/comments. 

Thank you for this observation, and we do apologize for the poor wording of this new text; we have reworded it as suggested, and the new writing is included here for your convenience:

A comparison of the intended (CAD) relative densities and those measured on LLS fabricated via FFF, and LCD is presented in Figure 15. Note that differences between the relative densities of CAD models and FFF samples are negligible. This is mainly due to the absence of material at the regions where there is a change of struts cross-section variation in adjacent unit cells (see figures 14a-14d). On the other hand, differences in relative density between CAD models and LCD samples are more notorious due to the combination of aspects. Firstly, LCD samples result in minimum defects at these regions where unit cells with different struts cross-section variations are adjacent (figure 14e); secondly, struts in LCD samples are thicker than the CAD model ones. It is also important to highlight that, among the CAD models and all the different strut cross-section variations, the maximum difference was 1.56%.
